# WIDE GRAPH NEURAL NETWORK

## ABSTRACT

Graph Neural Networks from the spatial and the spectral domains often suffer from the following problems: over-smoothing, poor flexibility, and low performance on heterophily. In this paper, we provide a unified view of GNNs from the matrix space analysis perspective to identify potential reasons for these problems and propose a new GNN framework to address them, called Wide Graph Neural Network (WGNN). We formulate GNNs as two components: one is for constructing a non-parametric feature space, and the other is for learning the parameters to re-weight the feature space. For instance, spatial GNNs encode the adjacency matrix multiplication as the feature space and stack layers to re-weight it, and spectral ones sum the polynomials to build the feature space and learn shared model weights. Instead, WGNN constructs the space by concatenating all polynomials and re-weights them individually. This mechanism reduces the unnecessary constraints on the feature space due to the concatenation, which avoids over-smoothing and allows independent parameters for better flexibility. Beyond the parameter independence property, WGNN enjoys further flexibility in adding matrices with arbitrary columns. For instance, by taking the principal components of the adjacency matrix, we can significantly improve the representation of heterophilic graphs. We provide a detailed theoretical analysis and conduct extensive experiments on eight datasets to show the superiority of the proposed WGNN. [1]

## 1 INTRODUCTION

Graph neural networks (GNNs) have demonstrated their great potential in representation learning for graph-structured data, such as social networks, transportation networks, protein interaction networks, and chemical structures (Fan et al., 2019; Wu et al., 2020; Zheng et al., 2022). Despite the success, existing GNNs still suffer some issues in the following. Firstly, the spatial GNNs aggregate the information from the connected nodes, resulting in the well-known over-smoothing (Cai & Wang, 2020). Secondly, the spatial models assume that the features of connected nodes are similar; however, this assumption does not hold in

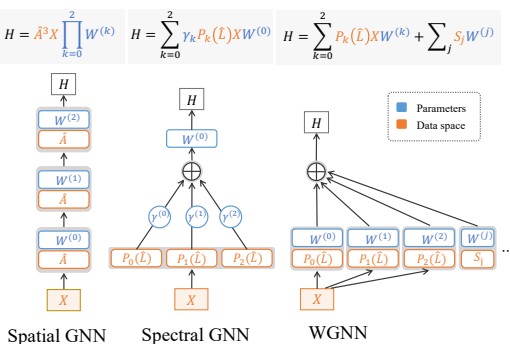

Figure 1: WGNN compared with current GNNs

heterophilic graphs (Zheng et al., 2022). Thirdly, the spectral GNNs use polynomials to approach arbitrary graph filters (He et al., 2021; Klicpera et al., 2019; Defferrard et al., 2016). In the absence of layer stacking, the spectral GNNs are exempt from the issue of over-smoothing. However, these spectral GNNs still perform poorly on heterophilic graphs since each polynomial term also shares the same assumption of similarity in neighbors. In addition, spectral methods share the parameters for each polynomial term, leading to a less flexible architecture. To better understand the problems in both spatial and spectral domains, efforts exist that integrate GNNs, e.g., from the perspective of optimization objectives (Ma et al., 2021; Zhu et al., 2021). However, they focus on summarizing general formulas while lacking a clear explanation of the problems.

---

[1]The implementation of WGNN is available at `https://drive.google.com/drive/folders/1A6VWiPmKRhCNfdcuFJvnxTiTgzgbJIZ6?usp=sharing`

In this paper, we propose a unified view for both spectral and spatial GNNs from the matrix space analysis point of view to investigate possible reasons for these problems and contribute a new way to address them. Specifically, for the sake of theoretical investigations, we first abstract a linear approximation of the GNNs following Wu et al. (2019a); Xu et al. (2018a). Then, as shown in the mathematical formulation and implementation structure of Figure 1, we decompose the components with and without parameters in the linear approximation, where the latter is regarded as a feature space built by node attributes and graph structure (e.g., adjacency or Laplacian matrices), and the former denotes the learnable parameters to re-weight the features. Consider spatial GNNs that 1) build the feature space by taking the power of the adjacency matrix, and 2) form the parameter space by taking the product of the weight matrices. For spectral GNNs, they sum the polynomials to compose the feature space and share the parameter for each. Based on this view, we can identify the reasons for issues in GNNs. When forming the feature space by powers of adjacency matrices, we find that over-smoothing is due to feature space compression. The parameter-sharing manner of spectral GNNs limits the flexibility of their architectures. Besides, the common issue of poor performance in heterophilic graphs is caused by the construction of each feature sub-space that embodies the similarity of neighboring nodes in both methods.

The primary contribution of this work is a wide architecture of GNNs named Wide Graph Neural Networks (WGNN), whose basic architecture is shown in Figure 1. In particular, it constructs the feature space by concatenating the polynomial terms of the adjacency matrix. This concatenation avoids space compression caused by powers in the spatial domain and alleviates the over-smoothing problem. To account for the feature space with multiple polynomial terms, the WGNN re-weights each one with an independent parameter matrix. Unlike spectral GNNs, which use a single parameter matrix for all polynomial terms, our WGNN has better flexibility by allowing different parameters for each. WGNN architectures also enjoy augmenting the feature space with arbitrary width of matrices. With this characteristic, we can improve the performance on heterophilic graphs by adding principal components of the adjacency matrix. This augmentation reduces the dependency of the feature space on the similarity of adjacent nodes since the principal components only extract the graph structure. Comprehensive experiments on both homophilic and heterophilic datasets demonstrate the superiority of WGNN.

**Contributions.** (1) We provide a unified view of both spatial and spectral GNNs, which formulates GNNs as the framework of jointly constructing the feature space and learning the parameters to re-weight. (2) We propose a new architecture, WGNN, which avoids over-smoothing, enjoys flexibility, alleviates heterophily problems, and provide a detailed theoretical analysis. (3) We conduct experiments on homophilic and heterophilic datasets and achieve significant improvements, e.g., an average accuracy increase of 32% on heterophilic graphs.

## 2 PRELIMINARIES

In this paper, we focus on the undirected graph $\mathcal{G} = (\mathcal{V}, \mathcal{E})$, along with its node attributes of $\mathcal{V}$ as $X \in \mathbb{R}^{n \times d}$ and adjacency matrix $A \in \mathbb{R}^{n \times n}$ to present $\mathcal{E}$. GNNs take the input of the node attributes and the adjacency matrix, and output the hidden node representations, as $H = \text{GNN}(X, A) \in \mathbb{R}^{n \times d}$. By default, we employ the cross-entropy loss function in the node classification task to minimize the difference between node label $Y$ and the obtained representation as $\mathcal{L}(H, Y) = -\sum_i Y_i \log \text{softmax}(H_i)$.

### 2.1 SPATIAL AND SPECTRAL GNNS

**Spatial GNNs** mostly fall into the message-passing paradigm. For any given node, it essentially aggregates features from its neighbors and updates the aggregated feature,

$$H_i^{(k+1)} = \sigma \left( \text{upd} \left( H_i^{(k)}, \text{agg} \left( \hat{A}_{ij}, H_j^{(k)}; j \in N(i) \right) \right) \right), \tag{1}$$

where $\sigma(\cdot)$ is a non-linear activation function, $H^{(k)}$ indicates the hidden representation in $k$-th layer, $\text{agg}$ and $\text{upd}$ are the aggregation and updating functions (Balcilar et al., 2021), $\hat{A} = (D + I)^{-1/2}(A + I)(D + I)^{-1/2}$ is the re-normalized adjacency matrix using the degree matrix $D$, and $N(\cdot)$ denotes the 1-hop neighbors.

Here, we provide two examples to specify this general expression. One is the vanilla GCN (Kipf & Welling, 2017) that adopts the mean-aggregation and the average-update, as shown in the left part of Figure 1. Its formulation is:

$$H^{(k+1)} = \sigma\left(\hat{A}H^{(k)}W^{(k)}\right). \tag{2}$$

The second example shows a different update scheme with skip-connection (Xu et al., 2018a; Li et al., 2019; Chen et al., 2020b), which is defined as follows,

$$H^{(k+1)} = \sigma\left(\alpha^{(k)}H^{(0)}W_0^{(k)} + \hat{A}H^{(k)}W_1^{(k)}\right), \tag{3}$$

where $\alpha^{(k)}$ controls the weight of each layer's skip-connection, $W_0^{(k)}, W_1^{(k)}$ are the transformation weights for the initial layer and the previous one, respectively.

**Spectral GNNs** originally employ the Graph Fourier transforms to get filters (Chung & Graham, 1997), such as using the eigendecomposition of the Laplacian matrix: $\hat{L} = I - \hat{A} = U\Lambda U^T$. In recent years, methods of this type have focused more on approximating arbitrary global filters using polynomials (Wang & Zhang, 2022; Zhu & Koniusz, 2020; He et al., 2021), which has shown superior performance and is written as

$$H = \sum_{k=0}^{K} \gamma^{(k)} P_k(\hat{L})\sigma(XW_1)W_2, \tag{4}$$

where $P_k(\cdot)$ donates a polynomial's $k$-order term; $\gamma^{(k)}$ is the adaptive coefficients and $W_1, W_2$ are learnable parameters. In Figure 1, we replace $W_1W_2$ with $W^{(0)}$. Note that some instances of spectral filters are not included in this paper, such as Levie et al. (2018); Thanou et al. (2014).

## 2.2 CHALLENGING ISSUES

**Over-smoothing:** In spatial GNNs, when stacking layers deep enough, the representations of connected nodes tend to be the same. Unlike deep models with tens of layers, GNNs often have only a few layers, for which there exists relievers such as DropEdge Rong et al. (2020) and the skip-connection scheme (Li et al., 2019; Xu et al., 2018b;a; Chen et al., 2020b) while having limited effect. Recent research shows that over-smoothing is the result of low-pass filters from a spectral perspective (Wu et al., 2019b; He et al., 2021).

**Homophily and heterophily:** Homophily and heterophily are the concepts of differentiating whether connected nodes share the same labels. According to the definition of $h = |\{Y_i = Y_j; (i,j) \in \mathcal{E}\}|/|\mathcal{E}|$ (Zhu et al., 2020), we consider a graph with a larger $h$ as more likely to be homophilic, otherwise heterophilic. GNNs were designed for homophilic graphs, making them unable to deal with heterophilic ones, sometimes even worse than MLPs (Zheng et al., 2022).

**Poor flexibility:** In spectral GNNs, all polynomial terms share the same parameter matrix due to the concentration on updating the coefficients only, as shown in Figure 1. This learning mechanism, unlike spatial GNNs, can form another weight matrix to the next feature subspace by layer-wise multiplications, resulting in the feature matrices of spectral methods being linearly correlated.

## 3 METHODS

We propose a unified view with a decomposition of the feature space and the parameters in GNNs. The primary motivation for the view is we consider the potential connections among the issues is how they use the graph data, i.e., the construction of the feature space. To conduct theoretical investigations of the feature space, we abstract a linear approximation of GNNs based on the success of linearization attempts of Wu et al. (2019a); Xu et al. (2018a). Specifically, we offer an overall formulation of linear approximation of arbitrary graph neural networks. $\overline{\text{GNN}}(X, \hat{A})$ as:

$$H = \overline{\text{GNN}}(X, \hat{A}) = \sum_{t=0}^{T-1} \Phi_t(X, \hat{A})\Theta_t, \tag{5}$$

where $\Phi_t(X, \hat{A}) \in \mathbb{R}^{n \times d_t}$ is the non-parametric feature space constructing function that inputs the graph data (e.g., node attributes and graph structure) and outputs a feature subspace, $\Theta \in \mathbb{R}^{d_t \times c}$ is the parameter space to re-weight the corresponding feature subspace for each class $c$, and $T$ is a hyper-parameter of the number of the feature sub-spaces that the GNN contains. In general, in this linear approximation, a GNN model forms $K$ feature sub-spaces, i.e., $\Phi_t$, and outputs the addition of all the re-weighted sub-spaces using the respective parameters $\Theta_t$. Note that the (total) feature space is the union of the sub-spaces as $\Phi = \{\Phi_t\}_{t=0,1,\cdots,T-1}$. Similarly, we have the (total) parameters

$\Theta = \{\Theta_t\}_{t=0,1,\cdots,T-1}$. Besides, the number of the subspaces $T$ that a GNN model obtains is not parallel with its layer/order, for which we will provide some examples in Section 3.1.

In what follows, we will first identify the feature space $\Phi$ and the parameters $\Theta$ for the existing GNNs. Then, leveraging the linear approximation, we will introduce our proposed wide-form GNN architecture called WGNN. Lastly, we will theoretically analyze the reasons behind the failures of existing GNNs, e.g., over-smoothing and poor performance of heterophily, and conclude the superiority of WGNN.

### 3.1 REVISITING SPATIAL AND SPECTRAL GNNs

**Spatial GNNs.**    We first transform the recursive formula of spatial GNNs, e.g., equation 1, to an explicit formula, by iterating from the initial node attributes that $H^{(0)} = X$ and ignoring the activation function. Following Section 2, we consider two examples of spatial GNNs: vanilla GCN (Kipf & Welling, 2017) and the one with skip-connections (Xu et al., 2018a).

The linear approximated explicit formula of a $K$-layer GCN is written as:

$$H^{(K)} = \hat{A}^K X \prod_{i=0}^{K-1} W^{(i)}, \tag{6}$$

which forms single feature space $\Phi_0 = \hat{A}^K X$ and parameters $\Theta_0 = \prod_{i=0}^{K-1} W^{(i)}$ with $T = 1$. While equation 3 furthermore considers skip-connections, whose $K$-layer linear approximated explicit formula is formualted as:

$$H^{(K)} = \left( \sum_{i=0}^{K-1} \hat{A}^i X \alpha^{(K-1-i)} W_0^{(K-1-i)} \prod_{j=K-i}^{K-1} W_1^{(j)} \right) + \hat{A}^K X \prod_{h=0}^{K-1} W_1^{(h)}. \tag{7}$$

By this decomposition, this GCN with skip-connections consists of $T = K+1$ feature sub-spaces. It forms each feature sub-space as $\Phi_t = \hat{A}^t X$. For the first $T-1$ sub-spaces, the according respective parameters is denoted as $\Theta_{t;t<T-1} = \alpha^{(K-1-t)} W_0^{(K-1-t)} \prod_{j=K-t}^{T-1} W_1^{(t)}$, and for for the last $\Phi_T$, the parameter is $\Theta_T = \prod_{h=0}^{T-1} W_1^{(h)}$. Please refer to the appendix A.1 for the derivation.

**Spectral GNNs.**    Spectral GNNs are specified by the explicit formula as equation 4. We remove the activation function, and obtain the linear approximation of a $K$-order spectral GNNs as:

$$H^{(K)} = \sum_{k=0}^{K} P_k(\hat{L}) X \gamma^{(k)} W^{(0)}. \tag{8}$$

We put the learnable polynomial coefficient $\gamma^{(k)}$ together with the parameter matrices. Also, we combine the shared parametric matrices in equation 4 as $W^{(0)} = W_1 W_2$. In this way, equation 8 forms $T = K + 1$ feature sub-spaces, where each sub-space is denoted as $\Phi_t = P_t(\hat{L})X$, and the parameters utilized to re-weight the respective sub-spaces are $\Theta_t = W^{(0)} W^{(1)}$.

**Primary analysis.**    In Table 1, we summarize more instances of spatial and spectral methods, with different colors to distinguish the feature space $\Phi$ (orange) and parameters $\Theta$ (blue). It demonstrates that the proposed uniform view can support most of the methods in both spatial and spectral domains. Due to the page limits, we put the example of GCNIIChen et al. (2020b) and ARMA Bianchi et al. (2021) in Appendix C.8. Compared to the general formulation of re-weighting feature sub-spaces, e.g., equation 5, existing GNNs prohibit constraints on both the feature space and the parameter space. We can observe that the feature space $\Phi$ of spatial GNNs is always constrained by the power of the adjacent matrix, which is potentially related to the over-smoothing problems. Most of the parameters $\Theta$ of spectral GNNs are shared for different sub-spaces, which limits the flexibility of adequately re-weighting for each sub-space. Besides, the feature space $\Phi$ in both spatial and spectral GNNs is formulated by multiplication of structural matrices function and node attributes (e.g., $\Phi_k = P_k(\hat{L})X$). This multiplication to the node attributes $X \in \mathbb{R}^{n \times d}$ demands the feature sub-spaces to obtain $d$ columns, which prevents the feature matrices with other shapes.

Table 1: The feature space and parameters of the linear approximation for GNN models.

| | Original formula* | Linear approximation formulations |
|---|---|---|
| GCN (Kipf & Welling, 2017) | $H^{(k+1)} = \sigma\left(\hat{A}H^{(k)}W^{(k)}\right)$ | $H^{(K)} = \hat{A}^K X \prod_{i=0}^{K-1} W^{(i)}$ |
| GIN (Xu et al., 2018a) | $H^{(k+1)} = \sigma\left((\epsilon^{(k)}I + \hat{A})H^{(k)}W_0^{(k)}\right)W_1^{(k)}$ | $H^{(K)} = \sum_{t=0}^{K} \hat{A}^k X \sum_{\{q_0, \cdots, q_{K-1}\} \subseteq \{\epsilon^{(0)}, \cdots, \epsilon^{(K-1)}\}} \prod_i q_i \cdot \prod_{j=0}^{K-1} W_0^{(j)} W_1^{(j)}$ |
| APPNP (Klicpera et al., 2019) | $H^{(k+1)} = (1-\alpha)\hat{A}H^{(l)} + \alpha H^{(0)}; H^{(0)} = \sigma(XW_1)W_2$ | $H^{(K)} = \sum_{t=0}^{K} (1-\alpha)^t \hat{A}^t H^{(0)} + \sum_{i=0}^{t-1} \alpha(1-\alpha)^i \hat{A}^i H^{(0)} W_1 W_2$ |
| ChebyNet (Defferrard et al., 2016)** | $H = \sum_{k=0}^{K} P_k(\hat{L})XW^{(k)}$ | $H^{(K)} = \sum_{t=0}^{K} P_t(\hat{L})XW^{(t)}$ |
| GPRGNN (Chien et al., 2021) | $H = \sum_{k=0}^{K} \gamma^{(k)} \hat{L}^k \sigma(XW_1)W_2$ | $H^{(K)} = \sum_{t=0}^{K} \hat{L}^t X \gamma^{(t)} W_1 W_2$ |
| BernNet (He et al., 2021) | $H = \sum_{k=0}^{K} \frac{1}{2^K}\binom{K}{k}\gamma^{(k)}(2I-\hat{L})^{K-k}\hat{L}^k\sigma(XW_1)W_2$ | $H^{(K)} = \sum_{t=0}^{K} (2I-\hat{L})^{K-l}\hat{L}^t X \gamma^{(t)} W_1 W_2$ |
| WGNN (Ours) | $H = \sum_{k=0}^{K} P_k(\hat{L})XW^{(k)} + SW^{(s)}$ | $H = \sum_{k=0}^{K-1} P_k(\hat{L})XW^{(l)} + \sum_{j=0}^{J-1} S_j W^{(j)}$ |

* Without specification, $H^{(0)} = X$; ** $T_k(x)$ denotes Chebyshev polynomial $P_0(x) = 1, P_1(x) = x, P_k(x) = 2xP_{k-1} - P_{k-2}$.

## 3.2 OUR PROPOSAL: WIDE GRAPH NEURAL NETWORK

Given the observations in the last part, we propose a Wide Graph Neural Network, a generalized framework of GNNs that relaxes the constraints as formulated in the following,

$$H = \sum_{k=0}^{K} P_k(\hat{L})XW^{(k)} + \sum_{j=0}^{J-1} S_j W^{(j)}. \tag{9}$$

It constructs the feature space in a two-fold way. The first part inherits the previous GNNs, that the same size of the feature sub-spaces is formed by the multiplication of the polynomials of the structural matrix $P_k(\hat{L})$ and the node attributes $X$. Secondly, we allow the feature sub-spaces $S_j$ with an arbitrary number of the columns, instead of the same columns with $X$. Beyond these, WGNN utilizes independent parameters matrices $W^{(k)}$ and $W^{(j)}$ to re-weight each feature sub-spaces to provide flexible re-weighting. To sum up, WGNN forms the feature space of $T = K + J$ sub-spaces, denoted as $\Phi_k = P_k(\hat{L})X \in \mathbb{R}^{n \times d}$ similar to GNNs', and $\Phi_j = S_j \in \mathbb{R}^{n \times d_j}$, the additional part with arbitrary columns ones, which compose the total feature space $\Phi = \{\Phi_k\}_{k=0,1,\cdots,K-1} \cup \{\Phi_j\}_{j=0,1,\cdots,J-1}$. Respectively, $\Theta_k \in \mathbb{R}^{d \times c}$ and $\Theta_j \in \mathbb{R}^{d_j \times c}$ re-weight them with respect to the objective.

In general, $S_j$ could be any transformation of node features $X$, graph structure $\hat{A}$, or both of them. The feature spaces $\Phi_k = P_k(\hat{L})X \in \mathbb{R}^{n \times d}$ provides the usage of the node features $X$ only, e.g. $k = 0$, and both of node attributes and graph structure, e.g. $k > 0$. Besides, using node features leads to the dependency of the adjacent nodes' similarity and is parallel to the heterophily problem. In WGNN, we break this dependency and form $S_j$ by using the graph structure only. To extract the low-dimension information for the graph structure, we deploy truncated SVD to get its principal components as follows:

$$S = \tilde{Q}\tilde{V}; \hat{A} = QVR^T, \tag{10}$$

where $S$ denotes that we only use single $S_j$, e.g., $J = 1$. Throughout the remaining context, we stick to truncated SVD as a case for WGNN and delve into it accordingly. In addtion, the empirical results of other transformation functions are given in Appendix C.4 Notably, the feature space $\Phi_k$ and $S$ may have imbalanced scales and cause poor re-weighting. We, therefore, add a column-wise normalization to ensure each column equally contributes to the whole feature space.

## 3.3 THEORETICAL ANALYSIS

In this part, we analyze the feature space that different GNNs formed to explain the challenges of over-smoothing and poor performance on heterophilic graphs.

**Over-smoothing.** The over-smoothing problem occurs when stacking deep GNN layers. We describe this phenomenon by the compression of the column span of the feature space. It is defined as all the possible column-wise linear combinations of the matrix's columns and denoted as $\mathrm{Span}(\Phi) = \{\sum_i a_i \Phi_{\cdot i}; a_i \in \mathbb{R}\}$. We provide Theorem 3.1 to interpret the cause of this issue.

**Theorem 3.1.** *The span of the feature space $\Phi_k = \hat{A}^k X$ will be shrunk gradually with the increase of $k$, which leads to the over-smoothing problem.*

Let us first look at the feature space of $K$-layer vanilla GCN,

$$\hat{A}^K X = (I - \hat{L})^K X = U(I - \Lambda)^l U^T X; \Lambda \in [0, 2]^n.$$

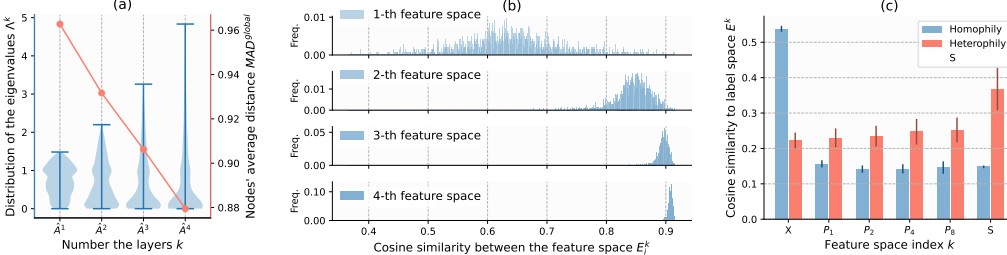

Figure 2: Visualization of the analysis. (a) collects the distribution of the eigenvalues with increasing powers of the adjacency matrix. It shifts to zero gradually, which compresses the feature space and leads to decreasing the diversity of all the nodes' features, measured by $\mathtt{MAD}^{\mathtt{global}}$ (Chen et al., 2020a). (b) shows the similarity between the later feature space to the previous total improves greatly, leading to marginal extension of the space. (c) compares the similarity of different feature spaces' distances to the labels, where the heterophilic case obtains a greater distance and our proposed $S$ can reduce the distance.

It is a linear combination of $U(I - \Lambda)^l$ using $U^T X$, and $(I - \Lambda)^l$ re-weights $U$ in column-wise. Since $\Lambda \in [0, 2]^2$, we have $(I - \Lambda) \in [-1, 1]^n$. Along with the increase of $K$, the weights of some $U$' columns will approach zero to shrink $\mathrm{Span}(U(I - \Lambda)^l)$, because

$$\mathrm{Span}(\hat{A}^K X) = \mathrm{Span}(U(I - \Lambda)^K U^T X) \subset \mathrm{Span}(U(I - \Lambda)^K).$$

Figure 2(a) demonstrates this phenomenon since the distribution of the eigenvalues shifts to zero with increasing layers while the similarity of the combination $\hat{A}^K X$ greatly enhances. In this way, over-smoothing occurs due to the limited column span of the feature space that compresses the representations. To alleviate this issue, some modified GNNs using skip-connections that form the feature space as $\{\hat{A}^k X\}_{k=0,1,\cdots,K}$ that joints successive layer's spaces, that

$$\mathrm{Span}(\{\hat{A}^k X\}_{k=0,1,\cdots,K}) = \mathrm{Span}(\hat{A}^0 X) \cup \mathrm{Span}(\hat{A}^1 X) \cup \cdots \cup \mathrm{Span}(\hat{A}^K X).$$

It will not be influenced by the compression of later components, e.g., $\mathrm{Span}(\hat{A}^K X)$, and therefore avoids over-smoothing. Similar conclusion can be derived by the feature space of spectral type that forms $\{\hat{L}^k X\}_{k=0,1,\cdots,K}$. Following this, we can understand the performance bottleneck with increasing layers by the similarity of the introduced feature space from later spatial layers / spectral orders to the previous ones. Here, we provide quantitative analysis to better describe the property. For this purpose, we take the feature space of spectral GNN (Chien et al., 2021) as an example, i.e., $\{\hat{L}^k X\}_{k=0,1,\cdots,4}$, and measure the linear correlation of the appended $k$-th feature space to the previous ones by calculating the mutual-correlation values:

$$E_i^k = \max_{j=0,\cdots,k-1} \mu(\hat{L}^j X, \hat{L}^k X_{\cdot i}),$$

where $i$ is the index of the column in $\hat{L}^k X$, and $\mu(M_0, M_1) = \max_{d_u \in M_0, d_v \in M_1} \cos(d_u, d_v)$ is the mutual-coherence of two matrices, based on the cosine distance $\cos$. In Figure 2(b), we visualize the distribution of $\{E_i^k\}$ of all the columns with $k = 1, 2, 3, 4$. It confirms the great improvement of the linear correlation, which results in little expansion of the feature space. Therefore, increasing spectral orders or spatial layers can hardly enhance performance. Although some studies explain the over-smoothing problem, e.g., Huang et al. (2020); Oono & Suzuki (2019); Cai & Wang (2020). Our perspective differs from them in the view concept; please refer to the comparison we provide in Appendix B.3.

**Poor performance in heterophily.** The majority of GNNs' performance on heterophilic graphs is much worse than on homophilic graphs. To understand this issue from the perspective of matrix space analysis, we study the linear correlation of the feature space to the label space. We provide an empirical analysis of the distribution following mutual-coherence values in Figure 2(c),

$$E^k = \frac{1}{C} \sum_c E_c^k; E_c^k = \mu(\hat{L}^k X_{\mathcal{T}\cdot}, Y'_{\mathcal{T}c})$$

where we randomly sample 60% rows to mimic the training set denoted as $\mathcal{T}$, and report the mean with variance. $c$ is the dimension of the matrix of the one-hot node labels, i.e., $Y' \in \mathbb{R}^{n \times c}$. It

shows the distance between the feature space and the label space on homophilic graphs is much higher than in the heterophilic scenario, which leads to poorer performance in the heterophily case. Besides, we append a theoretical explanation from the feature space only, by Theorem A.1, that *the mutual-coherence of the heterophilic feature space, i.e., $\mu(\hat{L}X)$, is higher than the homophilic ones.*

**WGNN compared to spatial and spectral GNNs.** Our WGNN provides a new way of dealing with graph data, where both graph structures and node attributes are regarded as the input cues to construct feature spaces. In this way, the complex multiplicative design between the graph structure and node attributes can be avoided and the constraints of feature space are relaxed, which contributes to better model flexibility and generalizability on the heterophilic graph.

From the feature space construction perspective, WGNN utilizes all the sub-spaces from $P_k(\hat{X})$. Compared with the spatial GNNs, the feature space of our WGNN is more flexible as $\text{Span}(\{\hat{A}^k X\}_{k=0,1,\cdots,K}) \subset \text{Span}(\{P_k(\hat{L})X_{k=0,1,\cdots,K})$, since it builds the space using all the polynomial terms while spatial GNNs only takes the highest ordered term. This property helps WGNN avoid the over-smoothing problem. Besides, we append a sub-space $S$ to the feature space of WGNN, which is built with the graph structure only. Without using the node attributes, this sub-space is closer to the label space than others that are highly dependent on nodes' similarity, and achieve better performance on the heterophilic graphs. As shown in Figure 2(c), we demonstrate that the sub-space $S$ helps the feature space to approach labels, especially for heterophilic graphs.

From the view of parameters, compared with spectral GNNs, WGNN relaxes all the constraints on the parameters and allows to re-weight the feature space independently. In Appendix A.4, we supply the demonstration of the parameters constraints within previous GNNs. It tells that the constraints on parameters ($W^{(k)}$) limit the span of the weighted feature space (see Theorem A.2).

# 4 EXPERIMENTS

We evaluate the proposed WGNN on the following aspects: (1) node classification results, (2) robustness on the challenging issues, (3) ablation studies.

**Dataset.** We implement our experiments on homophilic datsets, i.e., Cora, CiteSeer, PubMed, Computers, and Photo (Yang et al., 2016; Shchur et al., 2018), and heterophilic Chameleon, Squirrel and Actor (Rozemberczki et al., 2021; Pei et al., 2020). More details are provided in Appendix C.

**Baselines.** We compare a list of state-of-the-art GNN methods. For spatial GNNs, we have GCN (Kipf & Welling, 2017), GAT (Velickovic et al., 2018), GraphSAGE (Hamilton et al., 2017), GCNII (Chen et al., 2020b) and APPNP (Klicpera et al., 2019), where MLP is included as a particular case. For spectrals, we take ChebyNet (Defferrard et al., 2016), GPRGNN (Chien et al., 2021) and BernNet (He et al., 2021). Besides, we cover the recent unified models, ADA-UGNN (Ma et al., 2021) and GNN-LF/HF (Zhu et al., 2021). WGNN employs the Chebyshev or Monomial polynomials to construct feature space, and we name the corresponding version as WGNN-C and WGNN-M, respectively. Please refer to Appendix C.3 for more details about the implementation.

Table 2: Overall Performance of Wide Graph Neural Networks (WGNN)

| Type | Baseline | Time (ms) | Homophilic graphs | | | | | Heterophilic graphs | | |
|---|---|---|---|---|---|---|---|---|---|---|
| | | | Cora | CiteSeer | PubMed | Computers | Photo | Squirrel | Chameleon | Actor |
| Spatial | MLP | - | $76.70_{\pm0.15}$ | $76.67_{\pm0.26}$ | $85.11_{\pm0.26}$ | $82.62_{\pm0.21}$ | $84.16_{\pm0.13}$ | $37.86_{\pm0.39}$ | $57.83_{\pm0.31}$ | $38.99_{\pm0.17}$ |
| | GCN | $17.42_{\pm1.64}$ | $87.69_{\pm0.40}$ | $79.31_{\pm0.46}$ | $86.71_{\pm0.18}$ | $83.24_{\pm0.11}$ | $88.61_{\pm0.36}$ | $47.21_{\pm0.59}$ | $61.85_{\pm0.38}$ | $28.61_{\pm0.39}$ |
| | GAT | $18.06_{\pm1.18}$ | $88.07_{\pm0.41}$ | $80.80_{\pm0.26}$ | $86.69_{\pm0.14}$ | $82.86_{\pm0.35}$ | $90.84_{\pm0.32}$ | $33.40_{\pm0.16}$ | $51.82_{\pm1.33}$ | $33.48_{\pm0.35}$ |
| | GraphSAGE | $10.72_{\pm0.25}$ | $87.74_{\pm0.41}$ | $79.20_{\pm0.42}$ | $87.65_{\pm0.14}$ | $87.38_{\pm0.15}$ | $93.59_{\pm0.13}$ | $48.15_{\pm0.45}$ | $62.45_{\pm0.48}$ | $36.39_{\pm0.35}$ |
| | GCNII | $8.48_{\pm0.24}$ | $87.46_{\pm0.31}$ | $80.76_{\pm0.30}$ | $88.82_{\pm0.08}$ | $84.75_{\pm0.22}$ | $93.21_{\pm0.25}$ | $43.28_{\pm0.35}$ | $61.80_{\pm0.44}$ | $38.61_{\pm0.26}$ |
| | APPNP | $23.74_{\pm2.08}$ | $87.92_{\pm0.20}$ | $81.42_{\pm0.26}$ | $88.16_{\pm0.14}$ | $85.88_{\pm0.13}$ | $90.40_{\pm0.34}$ | $39.63_{\pm0.49}$ | $59.01_{\pm0.48}$ | $39.90_{\pm0.25}$ |
| Spectral | ChebyNet | $20.26_{\pm1.03}$ | $87.17_{\pm0.19}$ | $77.97_{\pm0.36}$ | $89.04_{\pm0.08}$ | $87.92_{\pm0.13}$ | $94.58_{\pm0.11}$ | $44.55_{\pm0.28}$ | $64.06_{\pm0.47}$ | $25.55_{\pm1.67}$ |
| | GPRGNN | $23.55_{\pm1.26}$ | $87.97_{\pm0.24}$ | $78.57_{\pm0.31}$ | $89.11_{\pm0.08}$ | $86.07_{\pm0.14}$ | $93.99_{\pm0.11}$ | $43.66_{\pm0.22}$ | $63.67_{\pm0.34}$ | $36.93_{\pm0.26}$ |
| | BernNet | $36.88_{\pm0.84}$ | $87.66_{\pm0.26}$ | $79.34_{\pm0.32}$ | $89.33_{\pm0.07}$ | $88.66_{\pm0.08}$ | $94.03_{\pm0.08}$ | $44.57_{\pm0.33}$ | $63.07_{\pm0.43}$ | $36.89_{\pm0.30}$ |
| Unified | GNN-LF | $52.77_{\pm4.50}$ | $88.12_{\pm0.06}$ | $\mathbf{83.66_{\pm0.06}}$ | $87.79_{\pm0.05}$ | $87.63_{\pm0.05}$ | $93.79_{\pm0.06}$ | $39.03_{\pm0.08}$ | $59.84_{\pm0.09}$ | $41.97_{\pm0.06}$ |
| | GNN-HF | $53.28_{\pm4.51}$ | $88.47_{\pm0.09}$ | $83.56_{\pm0.10}$ | $87.83_{\pm0.10}$ | $86.94_{\pm0.06}$ | $93.89_{\pm0.10}$ | $39.01_{\pm0.51}$ | $63.90_{\pm0.11}$ | $\mathbf{42.47_{\pm0.07}}$ |
| | ADA-UGNN | $14.36_{\pm0.21}$ | $88.92_{\pm0.11}$ | $79.34_{\pm0.09}$ | $90.08_{\pm0.05}$ | $89.56_{\pm0.09}$ | $94.66_{\pm0.07}$ | $44.58_{\pm0.16}$ | $59.25_{\pm0.16}$ | $41.38_{\pm0.12}$ |
| | **WGNN-C** | $15.8_{\pm0.11}$ | $\mathbf{89.45_{\pm0.22}}$ | $81.96_{\pm0.23}$ | $\mathbf{90.27_{\pm0.49}}$ | $\mathbf{90.79_{\pm0.08}}$ | $95.36_{\pm0.14}$ | $67.82_{\pm0.26}$ | $\mathbf{73.33_{\pm0.35}}$ | $40.54_{\pm0.15}$ |
| | **WGNN-M** | $14.6_{\pm0.32}$ | $89.09_{\pm0.22}$ | $81.76_{\pm0.23}$ | $89.93_{\pm0.23}$ | $90.60_{\pm0.11}$ | $\mathbf{95.45_{\pm0.15}}$ | $\mathbf{67.90_{\pm0.23}}$ | $73.26_{\pm0.38}$ | $40.91_{\pm0.22}$ |

## 4.1 NODE CLASSIFICATION

We test on transductive node classification task with random 60%/20%/20% splits and summarize the results of 100 runs in Table 2, reporting the average accuracy with a 95% confidence interval. We observe that WGNN has almost the best performance on homophilic graphs. Particularly, compared with the current SoTA method ADA-UGNN (Ma et al., 2021) which unifies the objectives in both spatial and spectral domains, our WGNN achieves 1.1% accuracy improvement on average of 5 homophilic graphs datasets. Besides, WGNN obtains **32.0**% improvement on average of three heterophilic graphs datasets than the GCN baseline. The excellent performance on both homophilic and heterophilic graphs indicates the potential of WGNN.

## 4.2 ROBUSTNESS ON THE CHALLENGING ISSUES

**Over-smoothing.** We evaluate WGNN with different numbers of feature sub-space $K$ on the Cora (homophilic) and Chameleon datasets (heterophilic). Figure 5 indicates that the performance of WGNN will not drop as extending feature space. It is because our concatenating polynomials of adjacent matrices avoids the compression of feature space.

**Heterophilic graphs.** As shown in Table 2 and Section 4.1, WGNN greatly alleviates the problem of heterophily by supplementing the feature space with only the graph structure to reduce the dependency on nodes' similarity.

Table 3: Ablation study of the components in WGNN

|  | Cora | CiteSeer | PubMed | Squirrel | Chameleon |
|---|---|---|---|---|---|
| WGNN-C | **89.45**$_{\pm 0.22}$ | **81.96**$_{\pm 0.23}$ | 89.87$_{\pm 0.49}$ | 67.82$_{\pm 0.26}$ | **73.33**$_{\pm 0.35}$ |
| WGNN-M | 89.09$_{\pm 0.22}$ | 81.76$_{\pm 0.23}$ | 89.93$_{\pm 0.23}$ | **67.90**$_{\pm 0.23}$ | 73.26$_{\pm 0.38}$ |
| w/o norm | 86.23$_{\pm 1.43}$ | 79.32$_{\pm 0.59}$ | **90.27**$_{\pm 0.49}$ | 64.70$_{\pm 1.10}$ | 68.25$_{\pm 1.64}$ |
| w/o $S$ | 89.20$_{\pm 0.93}$ | 81.95$_{\pm 0.87}$ | 89.76$_{\pm 0.46}$ | 43.21$_{\pm 0.99}$ | 61.54$_{\pm 1.52}$ |
| w/o $P_k(\hat{L})X_{k>0}$ | 71.10$_{\pm 1.72}$ | 74.38$_{\pm 1.01}$ | 86.61$_{\pm 0.54}$ | 67.90$_{\pm 0.96}$ | 73.35$_{\pm 1.21}$ |
| w/o $P_k(\hat{L})X_{k=0}$ | 84.70$_{\pm 1.05}$ | 58.60$_{\pm 2.19}$ | 85.84$_{\pm 0.45}$ | 65.75$_{\pm 0.63}$ | 72.61$_{\pm 1.60}$ |

## 4.3 ABLATION STUDIES

In this subsection, we study the contribution of different components in WGNN and answer the following questions.

**How does each sub-space affect, e.g., $P_k(\hat{L})X$ and $S$? $P_k(\hat{L})X$ and $S$ respectively matter on homophilic and heterophilic graphs.** We evaluate WGNN on 5 datasets of both homophilic and heterophilic graphs in 3 different feature space constructions: including $w/o\ S$, $w/o\ P_k(\hat{L})X_{k=0}$, and $w/o\ P_k(\hat{L})X_{k>0}$, which respectively denote building the feature spaces without graph structure, without note attributes, and without the combination of them. In the ablation results of Table 3, we found that $w/o\ S$ works well on homophilic graphs but fails on heterophilic ones, while the other two work oppositely.

**Does the column-wise normalization matter? It matters when the node scale is not huge.** As shown in Table 3, we find column-wise normalization works well in most cases, except for PubMed. It may be because the large node scale of PubMed causes the tiny value of normalized feature space.

**On what ratio of the truncated SVD is adequate? 94%** We conduct an experiment using different ratios of singular vectors and values to construct $S$, i.e., the top $j$ singular values obtains

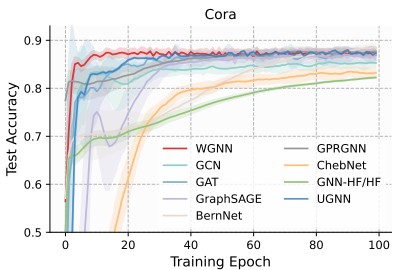

Figure 3: Convergence curve

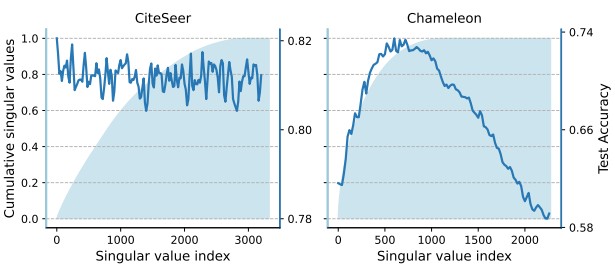

Figure 4: Sensitivity study of truncated SVD

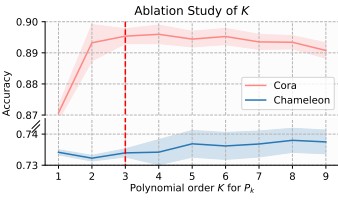
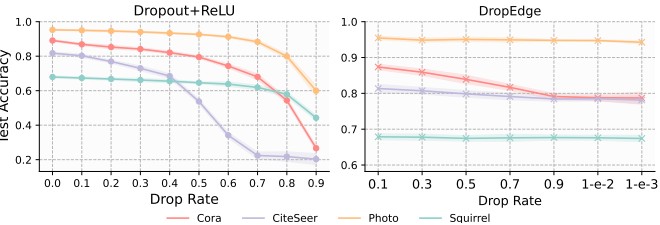

Figure 5: Ablation study of $K$

Figure 6: Analysis of the deep artifices on WGNN.

$r = \sum_{j=1}^{i} V_{jj} / \sum_{j=1}^{n} V_{jj}$ ratio of the components. In Figure 4, we report the test accuracy with respect to increasing ratios of singular values on CiteSeer and Chameleon, where CiteSeer is robust to the variation of the ratio, while Chameleon shows the best performance at $ratio = 94\%$. Thus, we use 94% for other experiments. We offer more interpretation of the SVD results in Appendix C.7.

**On what order polynomial is sufficient? Three.** We test a progressive order $K$ of the polynomials on Cora and Chameleon demonstrated in Figure 5. The performance in Cora rises from 1 to 3 and decreases in a slight tendency, while Chameleon has minor changes. It suggests that order 3 is good enough to achieve nearly optimal performance. Please refer to Appendix C.5 for more results.

**Are dropout (Agarap, 2018) and DropEdge (Rong et al., 2020) help? No.** We respectively integrate "dropout+ReLU" and DropEdge into WGNN and show the corresponding performance with different drop ratios on four datasets in Figure 6. Unfortunately, both of them show worse performance when increasing the drop rate. It may be because these regularization tricks break the graph structure. Besides, the results also advocate for more attention on the feature space construction, preventing over-applying deep artifices.

**Is WGNN easy to train? Comparably, yes, and WGNN converges fast.** In Table 2, we collect the training time per epoch (ms) for each method, which shows that WGNN behaves at a comparable time cost to other baselines, such as GCN (Kipf & Welling, 2017). Note that the time we report includes the graph propagation for a fair comparison, though WGNN can further reduce it by constructing the feature space in a pre-processing manner. This advantage comes from the lower-ordered polynomial feature space and much simpler computation in feed-forward. Please refer to Appendix C.3 for more details on the optimal architectures for the baselines. In Figure 3, we compare the convergence time for all methods and observe that WGNN consumes the minimum number of training epochs while achieving the highest accuracy.

**Is the SVD applicable in practice? Yes.** In Table 4, we show training time, SVD time (as preprocessing), and their ratio of WGNN. We find the rate of SVD time in whole training time is lower than 10%, which confirms the WGNN's applicability.

Table 4: Time consumption of SVD

|  | Cora | CiteSeer | Chameleon | Squirrel | Actor |
|---|---|---|---|---|---|
| Training time (ms) | $4000.01 \pm 52.23$ | $4103.46 \pm 133.77$ | $2818.21 \pm 81.87$ | $6096.43 \pm 403.95$ | $6074.39 \pm 547.34$ |
| SVD time (ms) | $3.88 \pm 0.08$ | $8.76 \pm 0.05$ | $61.80 \pm 0.24$ | $432.43 \pm 0.70$ | $3.99 \pm 0.02$ |
| # of epochs | 252 | 252 | 252 | 271 | 252 |
| SVD time / Training time (%) | 0.097 | 0.21 | 2.2 | 7.1 | 0.066 |

## 5 CONCLUSIONS, LIMITATIONS, AND FUTURE RESEARCH

In this paper, we provide a unified view to analyze GNNs, which separates the feature space and parameters using a linear approximation. Together, we provide a theoretical analysis of existing challenges under the setting of feature space or parameter space. To address these challenges, we propose a flexible architecture that relaxes all constraints, called Wide Graph Neural Network (WGNN). Comprehensive experiments are conducted to verify its superiority.

**Limitations and future research.** More general nonlinear cases are not included in our work, such as GAT (Velickovic et al., 2018), GateGNN (Bresson & Laurent, 2017), and will be considered in future work. The mechanism between the feature space and the respective parameters is worth more effort to optimize; in a way, the parameters in the WGNN can be further reduced by introducing reasonable constraints. Finally, since WGNN adopts the same graph structure as node attributes as graph data, more feature space construction methods should be discovered in the future.

## REPRODUCIBILITY STATEMENT

The code used in our experiments is provided in the supplementary material. For the data sets used in the experiments, a comprehensive description is given in Appendix C.

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

# Appendix

CONTENTS

# A PROOFS

## A.1 DERIVATION OF EQUATION 7

Iterate equation 3 from $H^{(0)} = X$, we have

$$H^{(0)} = X \tag{11}$$
$$H^{(1)} = \alpha^{(0)} X W_0^{(0)} + \hat{A} X W_1^{(0)} \tag{12}$$
$$H^{(2)} = \alpha^{(1)} X W_0^{(1)} + \hat{A} \alpha^{(0)} X W_0^{(0)} W_1^{(1)} + \hat{A}^2 X W_1^{(0)} W_1^{(1)} \tag{13}$$
$$H^{(3)} = \alpha^{(2)} X W_0^{(2)} + \hat{A} \alpha^{(1)} X W_0^{(1)} W_1^{(2)} \tag{14}$$
$$+ \hat{A}^2 \alpha^{(0)} X W_0^{(0)} W_1^{(1)} W_1^{(2)} + \hat{A}^3 X W_1^{(0)} W_1^{(1)} W_1^{(2)} \tag{15}$$
$$\cdots \tag{16}$$

Identify the rule of the iteration, we obtain

$$H^{(k)} = \sum_{i=0}^{l-1} \delta_i^{(k)} + \hat{A}^l X \prod_{h=0}^{l-1} W_1^{(h)}, \tag{17}$$

where $\delta_i^{(k)}$ s calculate by:

$$\delta_i^{(k)} = \alpha^{(k-1-i)} \hat{A}^i X W_0^{(k-1-i)} \prod_{j=l-i}^{l-1} W_1^{(j)}. \tag{18}$$

We apply equation 18 on equation 17 and put $\alpha^{(k-1-i)}$ back to the learnable parameters $W_0^{(k-1-i)}$, and the result of equation 7 is achieved.

## A.2 THE COLUMN-WISE NORMALIZATION IN CURRENT GNNS

Here, we include some 10-ordered polynomial functions, to see the different column-wise normalization response from these models. Column-wise normalization are defined as enforcing $\|F_{\cdot i}\|_2 = 1$, where we take $F$ as the concatenation of the feature space. We extend this to an arbitrary $k$ times of $\|F_{\cdot i}\|_2 = 1$, i.e., $\|F_{\cdot i}\|_2 = k$, which equals to measure the extent of the consistency of each $\|F_{\cdot i}\|_2$. Therefore, we report the standard variance of $\{\|F_{\cdot i}\|_2; i = 1, 2, \cdots\}$, and the smaller value suggests greater response of column-wise normalization.

Table 5: The column-wise normalization response for different polyonmials on Cora

|  | ChebyShev polynomial | Bernstein polynomial | Monomial polynomial |
|---|---|---|---|
| Cora | 3.8246 | 4.7044e-06 | 0.4947 |
| CiteSeer | 34.6432 | 0.0023 | 24.4430 |
| Chameleon | 660.4274 | 0.7308 | 1039.2469 |
| Squirrel | 245.6538 | 0.7063 | 700.9365 |

Chebshev, Bernstein and Monomial polynomials are compared in Table 5. Bernstein polynomial produces the least variance, suggesting it encourages the most atomicity compared to other polynomials. This observation aligns with the narrative in the original paper of BernNet He et al. (2021), where the authors claim the Bernstein polynomial is more numerically stable than other polynomial functions.

## A.3 EXPLAINING HETEROPHILY FROM THE PERSPECTIVE OF FEATURE SPACE

**Theorem A.1.** *The mutual-coherence of the heterophilic feature space, i.e., $\mu(LX)$, is higher than the homophilic ones.*

*Proof.* In a binary classification task, we assume the node features can be draw from two separate $p$-dimensional multivariate Gaussian distribution $\mathcal{D}_0 = \mathcal{N}(\vec{\mu_0}, \Sigma_0)$ and $\mathcal{D}_1 = \mathcal{N}(\vec{\mu_1}, \Sigma_1)$, corresponding to class $c_0$ and $c_1$. $\Sigma_0, \Sigma_1$ are both diagonal, i.e., for all $i, j; i \neq j$ dimensions are independent. The node features are equally sampled from the two distributions, $X = \{x_u; x_u \sim \mathcal{D}_0\} \cup \{x_v; x_v \sim \mathcal{D}_1\}$, where each includes $n$ samples. Without loss of generality, suppose two columns from $X$ that $d_i \perp d_j$. Equivalently, this leads to:

$$d_i^T d_j = \sum_{u \sim \mathcal{D}_0} x_{ui} x_{uj} + \sum_{v \sim \mathcal{D}_1} x_{vi} x_{vj} \tag{19}$$

$$\sim n\mathrm{E}(\mathcal{D}_{0i}\mathcal{D}_{0j}) + n\mathrm{E}(\mathcal{D}_{1i}\mathcal{D}_{1j}) \tag{20}$$

$$= n\mathrm{E}(\mathcal{D}_{0i})\mathrm{E}(\mathcal{D}_{0j}) + n\mathrm{E}(\mathcal{D}_{1i})\mathrm{E}(\mathcal{D}_{1j}) \tag{21}$$

$$= n(\mu_{0i}\mu_{0j} + \mu_{1i}\mu_{1j}) \tag{22}$$

$$= 0. \tag{23}$$

Note that for obtaining (64), we use the law of large numbers, e.g., $\sum_{i=0}^{n} A_i = \bar{A}$. So far, we have a equation $\mu_{0i}\mu_{0j} + \mu_{1i}\mu_{1j} = 0$ from this orthogonality, which is the only equation that our assumptions hold.

Next, we examine the effect of $L$ on $d_i^T d_j$. $L$ is employed by a left-hand side multiplication, which equals to a row-wise transformation of $d_i$ and $d_j$. We consider two extreme cases of homophily and heterophily, respectively.

Firstly, for homophily, $L$ only acts with the nodes that from the same class. For simplicity, if each transformation is averaged, i.e., $L_{ij}$ is row-wise normalized, $\mathcal{D}_0$ and $\mathcal{D}_1$ remain the same. Therefore, the orthogonal relation remains.

Secondly, $L$ combines the nodes that from different classes in heterophily. In this situation, $\mathcal{D}_0$ and $\mathcal{D}_1$ shift to $\mathcal{D}'_0 = \mathcal{N}(\vec{\mu_0} - \vec{\mu_1}, \Sigma_0 + \Sigma_1)$ and $\mathcal{D}'_1 = \mathcal{N}(\vec{\mu_1} - \vec{\mu_0}, \Sigma_0 + \Sigma_1)$. Here, we rewrite the alignment from (65):

$$d_i^T d_j = n\mathrm{E}(\mathcal{D}_{0i})\mathrm{E}(\mathcal{D}_{0j}) + n\mathrm{E}(\mathcal{D}_{1i})\mathrm{E}(\mathcal{D}_{1j}) \tag{24}$$

$$= n(\mu_{0i} - \mu_{1i})(\mu_{0j} - \mu_{1j}) + n(\mu_{1i} - \mu_{0i})(\mu_{1j} - \mu_{0j}) \tag{25}$$

$$= 2n(\mu_{0i} - \mu_{1i})(\mu_{0j} - \mu_{1j}) \tag{26}$$

$$= 2n(\mu_{0i}\mu_{0j} + \mu_{1i}\mu_{1j}) - 2n(\mu_{0i}\mu_{1j} + \mu_{1i}\mu_{0j}) \tag{27}$$

$$= -2n(\mu_{0i}\mu_{1j} + \mu_{1i}\mu_{0j}). \tag{28}$$

Based on this, the condition of orthogonality will not be held because $\mu_{0i}\mu_{1j} + \mu_{1i}\mu_{0j}$ is not equal to zero.

To sum up, heterophily breaks the limiting orthogonal condition, while homophily does not. Extensively, the $LX$ columns are more easily to be mixed in row-wise, losing their distinctiveness mutually. In other words, it increases the mutual-coherence of $LX$, for the definition below.

**Definition A.1.** *The mutual-coherence of a matrix $A \in \mathbb{R}^{n \times n}$ is the maximal inner product between columns from these two bases,*

$$\mu(A) = \max_{1 \leq i,j \leq n} |a_i^T a_j|, \tag{29}$$

*where each column is normalized as $\|a_i\|_2 = 1, 1 \leq i \leq n$.*

Based on this analysis and definition, we find that the mutual-coherence of the feature space in a heterophilic graph, e.g., $LX$, is more likely greater than that of a homophilic one. As a consequence, this phenomena will be superimposed in the overall feature space $F = \|\{\mathrm{P}_k(L)X; k = 0, 1 \cdots, K\}$ to undermine the power of the feature space.

$\square$

## A.4 PARAMETERS CONSTRAINTS IN PREVIOUS GNNs

**Theorem A.2.** *The span of the feature space $\Phi_k = \hat{A}^k X$ will be shrunk gradually with the increase of $k$, which leads to the over-smoothing problem.*

*Proof.* We summarize the constraints on $W$ in current GNNs as the following: i) in the case of MLP-based implementation He et al. (2020), all layers share the same $W$, which forces the layer-wise representation parameters into a single matrix; and ii) in the case of layer-wise $W$ Kipf & Welling (2017) Li et al. (2019), each $W_{k+1}$ is built upon its previous one, i.e, $W_{k+1} = \prod_{i=0}^{k+1} W_i$. We extract the ideas of these constraints into the following example. Suppose a linearly correlated feature space $U' = (d_0, d_1, \lambda_0 d_0, \lambda_1 d_1)$, where $d_0 \perp d_1, d_k \in \mathbb{R}^2$. $x \in \text{Span}\{d_0, d_1\}$ need to be recovered by the elements in $U'$.

We deploy the aforementioned two types of constraint on the undecided variables $b_0, b_1, b_2$, and $b_3$: i) $b_2 = b_0, b_3 = b_1$, and ii) $b_2 = \mu b_0, b_3 = \mu b_1$, where $\mu$ is a trainable scalar. They align with the graph neural networks. We begin by discussing these two cases.

Representing $x$ in the first case, yields:

$$x = b_0 d_0 + b_1 d_1 + b_0 \lambda_0 d_0 + b_1 \lambda_1 b_1 \tag{30}$$
$$= (1 + \lambda_0) b_0 d_0 + (1 + \lambda_1) b_1 d_1. \tag{31}$$

Using the unique representation theorem Hoffman & Kunze (2004), we have $(1 + \lambda_0) b_0 = a_0$ and $(1 + \lambda_1) b_1 = a_1$. Put it in a matrix multiplication format:

$$\begin{pmatrix} 1 & 0 & \lambda_0 & 0 \\ 0 & 1 & 0 & \lambda_1 \end{pmatrix} \begin{pmatrix} b_0 \\ b_1 \\ b_0 \\ b_1 \end{pmatrix} = \begin{pmatrix} a_0 \\ a_1 \end{pmatrix}, \tag{32}$$

which produces:

$$\begin{pmatrix} 1 + \lambda_0 & 0 \\ 0 & 1 + \lambda_1 \end{pmatrix} \begin{pmatrix} b_0 \\ b_1 \end{pmatrix} = \begin{pmatrix} a_0 \\ a_1 \end{pmatrix}. \tag{33}$$

It holds the closed form that $b_0 = \frac{a_0}{(1+\lambda_0)}, b_1 = \frac{a_1}{(1+\lambda_1)}$.

Then, we represent $x$ in the second case:

$$x = b_0 d_0 + b_1 d_1 + \mu b_0 \lambda_0 d_0 + \mu b_1 \lambda_1 b_1 \tag{34}$$
$$= (1 + \mu \lambda_0) b_0 d_0 + (1 + \mu \lambda_1) b_1 d_1, \tag{35}$$

which produces $(1 + \mu \lambda_0) b_0 = a_0$ and $(1 + \mu \lambda_1) b_1 = a_1$. Formulate them in a matrix multiplication:

$$\begin{pmatrix} 1 & 0 & \lambda_0 & 0 \\ 0 & 1 & 0 & \lambda_1 \end{pmatrix} \begin{pmatrix} b_0 \\ b_1 \\ \mu b_0 \\ \mu b_1 \end{pmatrix} = \begin{pmatrix} a_0 \\ a_1 \end{pmatrix}. \tag{36}$$

This is a under-determined system and gives $b_0 = \frac{a_0}{(1+\mu\lambda_0)}$, $b_1 = \frac{a_1}{(1+\mu\lambda_1)}$, $b_2 = \frac{\mu a_0}{(1+\mu\lambda_0)}$, and $b_3 = \frac{\mu a_1}{(1+\mu\lambda_1)}$.

We look into the values of $b_k$ to get the expressivity of the base $D$. Given the extreme case where $\lambda_0 \to 0$, the appended $\lambda_0 d_0$ is constrained while the original one keeps expressing. On the contrary, when $\lambda_0 \to \infty$, the original base $d_0$ is constrained by $\frac{a_0}{(1+\lambda_0)}$ or $\frac{a_0}{(1+\mu\gamma_0)}$ while the appended one expresses. Besides, for the second case, when $\mu \to 0$, the corresponding bases are limited by $b_2, b_3 \to 0$. Consequently, both cases lead to partial expression of the feature spaces.

Finally, comparing these two cases, i.e., (59) and (62) to (55), we find that they merely restrict the parameter space of $(b_0, b_1, b_2, b_3)^T$ by either sharing the values of each other or enforcing their linear dependence. Therefore, restricting the parameter space in these two cases leads to partial expression of the feature spaces. This proof is completed. □

In the following, we provide the norm of the learned parameter matrix respect to each column of the feature space, as shown in Figure 7. We find that existing GNNs' parameters are limited leaving a significant part of feature space unexplored. Differently, WGNN abandons all the constraints of the parameters and allows all the columns to be re-weighted.

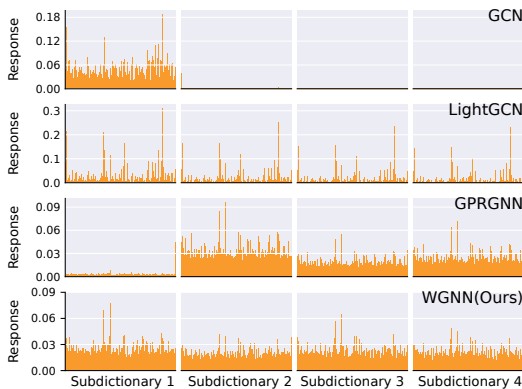

Figure 7: We present the expression of the feature space. Y-axis marks the respective norm of the parameters, i.e., $\|W_i\|_2$ of each $F_{\cdot i}$, and x-axis shows the index of the features spaces from different spatial layers or spectral orders. WGNN relaxes the constraint on the parameters, leading to a full expression on the feature space.

## B  RELATED WORK

### B.1  EFFORTS ON UNIFYING SEPARATE GNNS

Balcilar et al. (2021) first bridge the spatial methods to the spectral ones, that they assign most of the spatial GNNs with their corresponding graph filters. More specifically, they begin with GNN models' convolution matrix and then summarize their frequency responses. For example, GCN (Kipf & Welling, 2017) obtains the convolution matrix of $\tilde{D}^{-1/2}\tilde{A}\tilde{D}^{-1/2}$ leads to the filter of $\Phi_{GCN}(\lambda) \approx 1 - \lambda\bar{p}/(\bar{p}-1)$. This work causes attention to the unified perspective viewing GNNs, though they fail to explain the existing progress and issues in spectral view.

Ma et al. (2021) regard the aggregation progress of GCN (Kipf & Welling, 2017), GAT (Velickovic et al., 2018), and APPNP (Klicpera et al., 2019) as graph signal denoising problem, which aims to recover a clean signal $H$ from $\min_H \|H - X\|_F^2 + c \cdot \mathrm{tr}(H^T L H)$. Given this, the authors consider generalize the smoothing regularization term to $\sum_{i\in\mathcal{V}} C_i/2 \sum_{j\in\mathrm{N}(i)} \|H_i/\sqrt{d_i} - H_j/\sqrt{d_j}\|_2^2$ and propose ADA-UGNN. However, it also lacks the understanding of over-smoothing or heterophily.

Zhu et al. (2021) give a more comprehensive summary of GNNs from an optimization view, which partly overlaps with Ma et al. (2021)'s opinions of graph signal denoising. Based on their conclusion, they propose GNN-LF/HF with parameters adjusting the corresponding objective, e.g., GNN-LF approaches $\min_H \|I + \beta\tilde{L}^{1/2}(H-X)\|_F^2 + (1/\alpha - 1)\,\mathrm{tr}\!\left(H^T\tilde{L}H\right)$ and behaves as a low-pass filter. They attribute over-smoothing to the absence of original features and overcome this issue in their proposal; however, heterophily is untouched either.

In general, these integrated perspectives lack the explanation of the issues, but focus on general formulas.

### B.2  SHALLOW MODELS

SIGN (Rossi et al., 2020) practices to device the aggregation in a pre-processing way, which is similar to our idea of feature space construction. However, they focus on large-scale scenarios by removing sampling and aggregation operations from the training.

FSGNN (Maurya et al., 2021) implements feature selection on GNNs, similar to feature sub-space selection. Nevertheless, it combines all the sub-spaces by one linear transformation, which undermines the ability to understand the contribution of each part.

These shallow models empirically approaches to the idea of feature space, since they will not perform better on in deep models. Besides, concatenation is also an instance of the feature space concept, which is adopted by many spatial GNNs. In our view, these observations and modifications are are framed as a method to expand the feature space.

### B.3 OTHER ANALYTICAL VIEWS OF OVER-SMOOTHING

As we mentioned in Section 3, that several works exist explaining over-smoothing problems (including those you mentioned). However, the view concept of our perspective, i.e., matrix space analysis, is different from the existing ones. Oono & Suzuki (2019) propose one of the most accepted views, which is later followed by some literature Huang et al. (2020); Shan et al. (2021). They assumes a stable point of node feature made by the node degrees and proposes that whatever the initial node features are, some nodes will converge to the stable degree feature, with the layer going infinitely, which is depending on the node's corresponding eigenvalue. In details, the prove that $d_{\mathcal{M}}(X^{(l)}) \leq (s\lambda)^l d_{\mathcal{M}}(X^{(0)})$, where $d_{\mathcal{M}}(X^{(l)})$ is the distance to a given feature on the top of node-degree and converges to 0 when $s\lambda < 1$. For concise, let us call it degree-view. It analyzes from the row-wise perspective of the feature space. And some studies are also from a row-wise aspect by measuring the global Dirichlet energy of the node features Cai & Wang (2020).

Differently, our view starts from the column-wise of the feature space by comparing the (column) span of the feature space that extended. In particular, we factorize the feature space, for example, $L^k X$, as the bases (column-wise) composed by the eigenvector matrix $U$ (of $\hat{L}$). Each column is re-weighted by the corresponding eigenvalue $\Lambda_{ii}^k$, then expressed by the static weight matrix $U^T X$. Therefore, we consider the feature space spanned by the columns of $U\Lambda^k$ using the weight matrix $U^T X$, which is different from the node/row-wise perspective from the degree view.

Moreover, our view can potentially explain the inferior performance on heterophilic datasets than on homophilic ones. By Thm B.1, we can see the mutual coherence of the feature matrix (e.g., $\hat{L}X$) is higher in heterophilic settings, leading to a shrunk feature space.

## C EXPERIMENTAL SETTINGS

### C.1 DATASET DETAILS

The datasets are concluded in Table 6, with licenses. [2] [3] [4] Cora, CiteSeer, and PubMed are commonly used homophilic citation networks Yang et al. (2016). Computers and Photo are homophilic co-bought networks from Amazon Shchur et al. (2018). For heterophilic datasets, we utilize hyperlinked networks Squirrel and Chameleon from Pei et al. (2020), and Actor, a subgraph from the film-director-actor network Rozemberczki et al. (2021). PyG[5] are employed to get these data. Each datasets are split into three parts using random selection: 60% as the training set, 20% as the validation set, and 20% as the test set. We set these datasets to undirected graphs as we assumed in the Preliminaries.

Table 6: Statistics of Datasets

|  | Cora | CiteSeer | PubMed | Computers | Photo | Squirrel | Chameleon | Actor |
|---|---|---|---|---|---|---|---|---|
| $|\mathcal{V}|$ | 2,708 | 3,327 | 19,717 | 13,752 | 7,650 | 5,201 | 2,277 | 7,600 |
| $|\mathcal{E}|$ | 5,278 | 4,552 | 44,338 | 245,861 | 119,081 | 217,073 | 36,101 | 30,019 |
| $h(\mathcal{G})$ | 0.81 | 0.74 | 0.80 | 0.78 | 0.83 | 0.22 | 0.23 | 0.22 |
| $d(\mathcal{G})$ | 1.95 | 1.37 | 2.25 | 17.88 | 15.57 | 41.74 | 15.85 | 3.95 |

---

[2]Chameleon, Squirrel: https://github.com/benedekrozemberczki/MUSAE/blob/master/LICENSE

[3]Cora, CiteSeer, PubMed, Actor: https://networkrepository.com/policy.php

[4]Computers, Photo: https://github.com/shchur/gnn-benchmark/blob/master/LICENSE

[5]https://pytorch-geometric.readthedocs.io/en/latest/modules/datasets.html

## C.2 Details about optimization and reported results

We report the average accuracy (micro F1 score) in the classification task with a 95% confidence interval in all the tables and figures. For each result, we run 100 times on 10 random seeds. Besides, we present the standard variance of Table 2 in the Table 7 followed:

Table 7: Overall performance of WGNN compared to the baselines with reporting the standard variance.

| | Cora | CiteSeer | PubMed | Computers | Photo | Squirrel | Chameleon | Actor |
|---|---|---|---|---|---|---|---|---|
| MLP | $76.70_{\pm0.59}$ | $76.67_{\pm0.90}$ | $85.11_{\pm0.91}$ | $82.62_{\pm0.73}$ | $84.16_{\pm0.46}$ | $37.86_{\pm1.38}$ | $57.83_{\pm1.09}$ | $38.99_{\pm0.60}$ |
| GCN | $87.69_{\pm1.39}$ | $79.31_{\pm1.63}$ | $86.71_{\pm0.63}$ | $83.24_{\pm0.39}$ | $88.61_{\pm1.28}$ | $47.21_{\pm2.06}$ | $61.85_{\pm1.32}$ | $28.61_{\pm1.36}$ |
| GAT | $88.07_{\pm1.45}$ | $80.80_{\pm0.93}$ | $86.69_{\pm0.48}$ | $82.86_{\pm1.23}$ | $90.84_{\pm1.12}$ | $33.40_{\pm4.99}$ | $51.82_{\pm4.68}$ | $33.48_{\pm1.25}$ |
| GraphSAGE | $87.74_{\pm1.46}$ | $79.20_{\pm1.48}$ | $87.65_{\pm0.51}$ | $87.38_{\pm0.52}$ | $93.59_{\pm0.47}$ | $48.15_{\pm1.59}$ | $62.45_{\pm1.70}$ | $36.39_{\pm0.88}$ |
| GCNII | $87.46_{\pm0.18}$ | $80.76_{\pm1.05}$ | $88.82_{\pm0.75}$ | $84.75_{\pm0.78}$ | $93.21_{\pm0.89}$ | $43.28_{\pm1.21}$ | $61.80_{\pm1.54}$ | $38.61_{\pm0.90}$ |
| APPNP | $87.92_{\pm0.72}$ | $81.42_{\pm0.95}$ | $88.16_{\pm0.49}$ | $85.88_{\pm0.46}$ | $90.40_{\pm1.20}$ | $39.63_{\pm1.00}$ | $59.01_{\pm1.68}$ | $39.90_{\pm0.88}$ |
| ChebNet | $87.17_{\pm0.96}$ | $77.97_{\pm1.84}$ | $89.04_{\pm0.42}$ | $87.92_{\pm0.65}$ | $94.58_{\pm0.55}$ | $44.55_{\pm1.39}$ | $64.06_{\pm2.39}$ | $25.55_{\pm8.43}$ |
| GPRGNN | $87.97_{\pm1.23}$ | $78.57_{\pm1.56}$ | $89.11_{\pm0.44}$ | $86.07_{\pm0.71}$ | $93.99_{\pm0.54}$ | $43.66_{\pm1.12}$ | $63.67_{\pm1.72}$ | $36.93_{\pm1.30}$ |
| BernNet | $87.66_{\pm1.33}$ | $79.34_{\pm1.63}$ | $89.33_{\pm0.40}$ | $88.66_{\pm0.44}$ | $94.03_{\pm0.40}$ | $44.57_{\pm1.68}$ | $63.07_{\pm2.15}$ | $36.89_{\pm1.43}$ |
| GNN-LF | $88.12_{\pm0.06}$ | $\mathbf{83.66_{\pm0.06}}$ | $87.79_{\pm0.05}$ | $87.63_{\pm0.05}$ | $93.79_{\pm0.06}$ | $39.03_{\pm0.08}$ | $59.84_{\pm0.09}$ | $41.97_{\pm0.06}$ |
| GNN-HF | $88.47_{\pm0.09}$ | $83.56_{\pm0.10}$ | $87.83_{\pm0.10}$ | $86.94_{\pm0.06}$ | $93.89_{\pm0.10}$ | $39.01_{\pm0.51}$ | $63.90_{\pm0.11}$ | $\mathbf{42.47_{\pm0.07}}$ |
| ADA-UGNN | $88.92_{\pm0.11}$ | $79.34_{\pm0.09}$ | $90.08_{\pm0.05}$ | $89.56_{\pm0.09}$ | $94.66_{\pm0.07}$ | $44.58_{\pm0.16}$ | $59.25_{\pm0.16}$ | $41.38_{\pm0.12}$ |
| **WGNN-C** | $\mathbf{89.45_{\pm1.10}}$ | $81.96_{\pm1.18}$ | $89.87_{\pm0.74}$ | $\mathbf{90.79_{\pm0.40}}$ | $95.36_{\pm0.70}$ | $67.82_{\pm1.31}$ | $\mathbf{73.33_{\pm1.78}}$ | $40.54_{\pm0.79}$ |
| **WGNN-M** | $89.09_{\pm1.22}$ | $81.76_{\pm1.17}$ | $89.93_{\pm0.50}$ | $90.60_{\pm0.53}$ | $\mathbf{95.45_{\pm0.73}}$ | $\mathbf{67.90_{\pm1.18}}$ | $73.26_{\pm1.95}$ | $40.91_{\pm0.61}$ |
| w/o norm | $86.23_{\pm1.99}$ | $79.32_{\pm0.78}$ | $\mathbf{90.27_{\pm0.69}}$ | $89.43_{\pm0.55}$ | $94.94_{\pm0.78}$ | $64.70_{\pm1.53}$ | $68.25_{\pm2.30}$ | $37.46_{\pm0.86}$ |
| w/o $S$ | $89.20_{\pm1.30}$ | $81.95_{\pm1.16}$ | $89.76_{\pm0.65}$ | $89.10_{\pm0.60}$ | $94.56_{\pm0.77}$ | $43.21_{\pm1.39}$ | $61.54_{\pm2.12}$ | $40.89_{\pm0.60}$ |
| w/o $P_k$ | $71.10_{\pm2.41}$ | $74.38_{\pm20.86}$ | $86.61_{\pm0.75}$ | $89.58_{\pm0.56}$ | $94.90_{\pm0.44}$ | $67.90_{\pm1.34}$ | $73.35_{\pm1.70}$ | $38.44_{\pm1.01}$ |
| w/o $P_0$ | $84.70_{\pm1.47}$ | $58.60_{\pm3.06}$ | $85.84_{\pm0.64}$ | $90.02_{\pm0.32}$ | $92.92_{\pm0.72}$ | $65.75_{\pm0.88}$ | $72.61_{\pm2.23}$ | $25.89_{\pm4.80}$ |

We employ Adam for optimization and set the early stopping criteria as a warmup of 50 pluses patience of 200 for a maximum of 100 epochs. We conduct all the experiments on the machine with NVIDIA 3090 GPU (24G) and Intel(R) Xeon(R) Platinum 8260L CPU @ 2.30GHz.

## C.3 Searching space for baselines hyper-parameters

For WGNN, we turn the following hyper-parameters by the grid search.

- Learning rate: $\{0.01, 0.05, 0.1\}$
- Weight decay: $\{0.0005, 0.001, 0.005, 0.01, 0.02, 0.05\}$
- $|S|$ for homophilic graphs: $\{0, 10, 50, 100, 200, 500, 1000, 2000\}$
- $|S|$ for heterophilic graphs: $\{500, 600, 700, 800, 900, 1000, 1500, 2000\}$
- Suggested $|S|$: the whole hundred from the 94% singular values
- Hidden size: $64$
- Ranks $k$ of the polynomial $P_k(\hat{L})$: $\{0, 1, 2, 3\}$

Table 8: The universally used hyper-parameters for WGNN.

| | lr | weight decay | $|S|$ | hidden | $k$ |
|---|---|---|---|---|---|
| Cora | 0.01 | 0.01 | 50 | 64 | 3 |
| CiteSeer | 0.01 | 0.02 | 100 | 64 | 1 |
| PubMed | 0.01 | 0.005 | 100 | 64 | 3 |
| Computers | 0.01 | 0.0005 | 1000 | 64 | 3 |
| Photo | 0.01 | 0.0005 | 500 | 64 | 3 |
| Squirrel | 0.01 | 0.001 | 2000 | 64 | 3 |
| Chameleon | 0.01 | 0.0005 | 700 | 64 | 3 |
| Actor | 0.01 | 0.001 | 10 | 64 | 0 |

Table 9: The turned hyper-parameters for the baselines.

| | lr | weight decay | dropout | hidden | layers/ranks | others |
|---|---|---|---|---|---|---|
| MLP | $\{0.01, 0.05\}$ | 0.0005 | $\{0.5, 0.6, 0.8\}$ | 64 | 2 | - |
| GCN | $\{0.01, 0.05\}$ | 0.0005 | $\{0.5, 0.6, 0.8\}$ | 64 | $\{2,3\}$ | - |
| GAT | $\{0.01, 0.05\}$ | 0.0005 | $\{0.5, 0.6, 0.8\}$ | 64 | $\{2,3\}$ | $heads$:$\{1,8\}$ |
| GraphSAGE | $\{0.01, 0.05\}$ | 0.0005 | $\{0.5, 0.6, 0.8\}$ | 64 | $\{2,3\}$ | - |
| GCNII | $\{0.01, 0.05\}$ | 0.0005 | 0.5 | 64 | $\{2,4,10\}$ | $\alpha, \theta$:$\{0.1, 0.2, 0.5, 0.8, 0.9\}$ |
| APPNP | $\{0.01, 0.05\}$ | 0.0005 | 0.5 | 64 | $\{2,3,4,5,8\}$ | $\alpha$:$\{0.1, 0.2, 0.5, 0.8, 0.9\}$ |
| ChebNet | $\{0.005, 0.01, 0.05\}$ | $\{0.0, 0.0005\}$ | $\{0.1, 0.2, 0.5\}$ | 64 | 10 | - |
| GPRGNN | $\{0.005, 0.01, 0.05\}$ | $\{0.0, 0.0005\}$ | $\{0.1, 0.2, 0.5\}$ | 64 | 10 | - |
| BernNet | $\{0.005, 0.01, 0.05\}$ | $\{0.0, 0.0005\}$ | $\{0.1, 0.2, 0.5\}$ | 64 | 10 | $prop\_drate$:$\{0.001,0.02,0.01,0.05\}$ $prop\_lr$:$\{0.0, 0.1, 0.2, 0.5, 0.6, 0.7, 0.9\}$ |
| ADA-GNN | $\{0.05, 0.01\}$ | $\{0.0005, 0.00005\}$ | $\{0.2,0.5,0.8\}$ | 64 | $\{2,5,10\}$ | $s$:$\{1,9,19,29\}$ |
| GNN-LF | 0.01 | 0.0005 | 0.5 | 64 | 10 | $\alpha, \mu$: $\{0.1, 0.2, 0.3, 0.4, 0.5, 0.6, 0.8, 0.9\}$ |
| GNN-HF | 0.01 | 0.005 | 0.5 | 64 | 10 | $\alpha, \beta$: $\{0.1, 0.2, 0.3, 0.4, 0.5, 0.6, 0.8, 0.9\}$ |

Table 8 represents the hyper-parameters searched for the baselines used in our experiments. We prioritize their original released code repository, and the ranges of turning parameters are according to their papers.

- MLP, GCN, GAT GraphSAGE, APPNP, GCNII are implemented with PyG. [6]
- ChebNet is implemented according to the code style of BernNet/GPRGNN.
- GPRGNN is implemented according to its original code repository. [7]
- BernNet is implemented according to its original code repository. [8]
- ADA-UGNN is implemented according to its original code repository. [9]
- GNN-HF/LF are implemented according to its original code repository. [10]

## C.4 OTHER TRANSFORMATIONS FOR COMPACTING THE GRAPH STRUCTURE INFORMATION

We append other possible transformations to extract compacted information from the normalized adjacency matrix $\hat{A}$. In details, we compared:

- KernelPCA: a PCA method using non-linear kernel, where radial basis function (RBF) is used.
- FastICA: a fast version of independent components analysis, which is a linear method.
- IsoMap: a nonlinear dimensionality reduction method based on spectral theory.

All of them can be easily implemented by `sklearn` package. As shown in Table 10, our chosen truncated-SVD has comparable performance and we stick with this to make further analysis in the main text.

| | Cora | CiteSeer | Chameleon | Squirrel | Photo |
|---|---|---|---|---|---|
| None | $89.20 \pm 0.93$ | $81.95 \pm 0.87$ | $61.54 \pm 1.52$ | $43.21 \pm 0.99$ | - |
| Truncated-SVD (Ours) | $89.45 \pm 0.22$ | $81.96 \pm 0.23$ | $73.33 \pm 0.35$ | $67.90 \pm 0.23$ | $95.45 \pm 0.15$ |
| KernelPCA (non-linear) | $88.61 \pm 0.82$ | $81.99 \pm 1.11$ | $73.66 \pm 1.45$ | $68.79 \pm 1.13$ | $95.36 \pm 0.51$ |
| FastICA (linear) | $88.77 \pm 1.09$ | $81.92 \pm 1.00$ | $73.32 \pm 1.37$ | $68.12 \pm 0.97$ | $95.30 \pm 0.22$ |
| IsoMap (non-linear) | $88.54 \pm 0.86$ | $82.07 \pm 1.15$ | $67.00 \pm 1.54$ | $54.47 \pm 0.87$ | $94.88 \pm 0.34$ |

Table 10: Comparing the transformations in compacting the normalized adjacency matrix

---

[6]https://github.com/pyg-team/pytorch_geometric

[7]https://github.com/jianhao2016/GPRGNN

[8]https://github.com/ivam-he/BernNet

[9]https://github.com/alge24/ADA-UGNN

[10]https://github.com/zhumeiqiBUPT/GNN-LF-HF

## C.5 RESULTS OF WGNN USING DIFFERENT POLYNOMIAL ORDERS

In the main text, we implement the polynomial order $K$ within the range of three based on the empirical observations, e.g., Figure 5. Here, we provide more comprehensive results for different choices of $K$.

Table 11 indicates that we may find better $K$ in a wider range, while the improvement is possibly marginal.

| $K$ | 4 | 5 | 6 | 7 | 8 | 9 |
|---|---|---|---|---|---|---|
| Cora | $89.60 \pm 0.30$ | $89.44 \pm 0.25$ | $89.52 \pm 0.26$ | $89.35 \pm 0.24$ | $89.34 \pm 0.22$ | $89.08 \pm 0.25$ |
| CiteSeer | $80.66 \pm 1.09$ | $81.15 \pm 1.06$ | $81.11 \pm 0.89$ | $80.83 \pm 1.07$ | $80.54 \pm 1.03$ | $80.10 \pm 1.02$ |
| Computers | $91.01 \pm 0.46$ | $90.90 \pm 0.51$ | $90.98 \pm 0.42$ | $90.77 \pm 0.39$ | $90.82 \pm 0.41$ | $90.45 \pm 0.32$ |
| Chameleon | $73.42 \pm 0.40$ | $73.69 \pm 0.43$ | $73.62 \pm 0.43$ | $73.68 \pm 0.42$ | $73.80 \pm 0.39$ | $73.75 \pm 0.38$ |
| Squirrel | $68.26 \pm 0.78$ | $68.41 \pm 0.88$ | $68.55 \pm 0.82$ | $68.83 \pm 0.68$ | $68.92 \pm 0.76$ | $69.06 \pm 0.93$ |

Table 11: Comparison of different $K$ within 10

## C.6 A DEEPER STUDY OF OVER-SMOOTHING

In this section, we append a more concrete study of over-smoothing problem. Specifically, we test WGNN with $K = \{10, 20, 30, 40, 50, 60, 70, 80\}$ and compare with a representative method GCNII that overcomes over-smoothing. The results of Table 12 verify the effectiveness of WGNN on avoiding over-smoothing which expresses superior capability as Residual Connection and DropEdge and even achieve comparable results with GCNII. Note that all the hyper-parameters including Dropout, DropEdge, $\alpha, \theta$ for SkipConnection and GCNII is searched within $\{0.2, 0.5, 0.8\}$.

## C.7 MORE COMPREHENSIVE STUDY OF SVD

In the main test, we append the results on CiteSeer and Squirrel to better verify the importance of principal components of extracting information from adjacency matrix into $S_j$. As shown in the Figure 8 below, we find the precisely results as we shared in Section 4. Cora and CiteSeer both 1) have a more smoothing distribution of the singular values and 2) the information from graph structure is less important as the node features and their interaction, therefore, the change of the performance is more stable with introducing more principal components. On the other hand, Chameleon and Squirrel 1) have a more centralized distribution of the singular values and 2) graph structure is a more important information, resulting an tendency of the performance that increases first and then decrease. In general, we can achieve a satisfying results on both kind of datasets when 94% principal components are inclusive.

Here, we offer more intuition about using principal components. The principal components project and summarize a larger correlated variables into a smaller and more easily interpretable axes of variation. It is ideal for $S_j$ to embody the graph structure information from adjacency matrix, because the adjacency matrix is sparse and high-dimensional but each nodes are topologically correlated. However, the different components need to be distinct from each other to be interpretable otherwise they only represent random directions, which leads to noise.

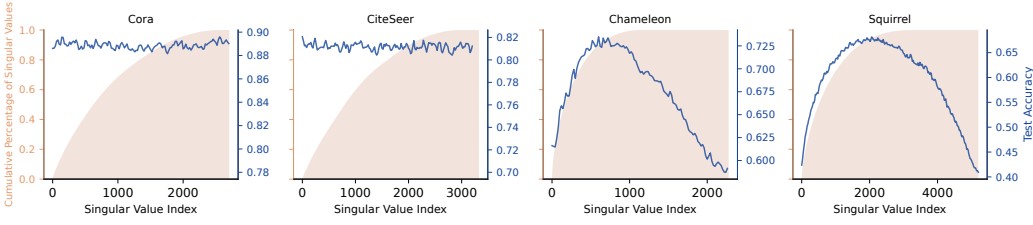

Figure 8: More results of SVD

Table 12: Studying Over-smoothing on WGNN, DropEdge, SkipConnection, and GCNII

| | K | 10 | 20 | 30 | 40 | 50 | 60 | 70 | 80 |
|---|---|---|---|---|---|---|---|---|---|
| Cora | WGNN | 88.92 ± 0.88 | 88.57 ± 1.75 | 88.60 ± 1.07 | 88.03 ± 0.69 | 87.98 ± 0.74 | 88.25 ± 0.58 | 87.93 ± 1.20 | 88.23 ± 0.97 |
| | DropEdge | 83.44 ± 1.83 | 77.38 ± 1.68 | 60.98 ± 2.18 | 55.93 ± 1.51 | 51.05 ± 0.58 | 50.91 ± 0.76 | 40.29 ± 2.40 | 36.45 ± 8.6 |
| | SkipConnection | 87.54 ± 0.56 | 87.03 ± 0.82 | 86.92 ± 0.47 | 86.89 ± 0.56 | 87.46 ± 0.66 | 87.00 ± 0.82 | 86.87 ± 0.62 | 86.89 ± 1.1 |
| | GCNII | 87.31 ± 1.44 | 87.80 ± 1.68 | 87.57 ± 2.18 | 88.09 ± 1.51 | 88.20 ± 1.44 | 88.52 ± 1.33 | 87.64 ± 1.87 | 88.16 ± 0.18 |
| Chameleon | WGNN | 74.02 ± 1.02 | 74.20 ± 1.46 | 74.01 ± 1.46 | 74.18 ± 1.22 | 74.18 ± 1.52 | 73.92 ± 1.39 | 73.84 ± 1.27 | 73.97 ± 1.18 |
| | DropEdge | 30.67 ± 2.03 | 22.71 ± 1.49 | 20.87 ± 1.18 | 21.87 ± 1.74 | 22.62 ± 1.35 | 21.72 ± 0.50 | 21.31 ± 3.99 | 21.51 ± 2.07 |
| | SkipConnection | 60.50 ± 0.45 | 59.71 ± 0.40 | 58.86 ± 0.53 | 58.16 ± 0.35 | 57.96 ± 0.40 | 58.89 ± 0.45 | 58.26 ± 0.52 | 58.53 ± 0.51 |
| | GCNII | 59.71 ± 0.71 | 58.86 ± 0.53 | 58.16 ± 0.35 | 57.96 ± 0.40 | 58.89 ± 0.45 | 58.26 ± 0.52 | 58.53 ± 0.51 | 58.53 ± 0.51 |
| Squirrel | WGNN | 68.44 ± 0.59 | 68.87 ± 0.82 | 68.70 ± 0.71 | 68.23 ± 1.08 | 68.13 ± 0.78 | 68.36 ± 0.85 | 67.62 ± 1.00 | 66.40 ± 1.20 |
| | DropEdge | 27.95 ± 0.98 | 28.42 ± 0.60 | 27.88 ± 0.81 | 26.56 ± 1.49 | 26.32 ± 1.26 | 23.92 ± 1.40 | 23.24 ± 1.60 | 22.40 ± 0.98 |
| | SkipConnection | 42.27 ± 0.50 | 41.33 ± 0.38 | 41.39 ± 0.39 | 40.11 ± 0.57 | 39.70 ± 0.39 | 40.25 ± 0.63 | 39.50 ± 0.54 | 40.06 ± 0.4 |
| | GCNII | 42.27 ± 0.50 | 41.33 ± 0.38 | 41.39 ± 0.39 | 40.11 ± 0.57 | 39.70 ± 0.39 | 40.25 ± 0.63 | 39.50 ± 0.54 | 40.06 ± 0.46 |

## C.8 FEATURE SPACE AND PARAMETERS FOR MORE GNN MODELS

Here, we present Table 1 in a more friendly way (with a larger scale and rotated 90 degrees), with adding GCNII Chen et al. (2020b).

Table 13: Feature Space and Parameters for More GNN Models

| | Original formula* | Linear approximation formulations |
|---|---|---|
| GCN (Kipf & Welling, 2017) | $H^{(k+1)} = \sigma\left(\hat{A}H^{(k)}W^{(k)}\right)$ | $H^{(K)} = \hat{A}^K X \prod_{i=0}^{K-1} W^{(i)}$ |
| GIN (Xu et al., 2018a) | $H^{(k+1)} = \sigma\left(\left(\epsilon^{(k)}I + \hat{A}\right)H^{(k)}W_0^{(k)}\right)W_1^{(k)}$ | $H^{(K)} = \sum_{k=0}^K \hat{A}^k X \sum_{\{q_0,\cdots,q_{K-t-1}\}\subseteq\{\epsilon^{(0)},\cdots,\epsilon^{(K-1)}\}} \prod_t q_t \cdot \prod_{j=0}^{K-1} W_0^{(j)}W_1^{(j)}$ |
| GCNII (Chen et al., 2020b) | $H^{(l+1)} = \sigma\left(\left(\left(1-\alpha^{(l)}\right)\hat{A}H^{(l)} + \alpha^{(l)}H^{(0)}\right)\left((1-\beta^{(l)})I + \beta^{(l)}W^{(l)}\right)\right)$ | $H^{(K)} = \sum_{t=0}^{K-1} \hat{A}^t X \prod_{i=L-t}^{L-1}(1-\alpha^{(i)})\alpha^{(L-t-1)}\prod_{j=L-t-1}^{L-1} W^{(j)} + \hat{A}^K \prod_{h=0}^{K-1}(1-\alpha^{(h)})W^{(h)}$ |
| ARMA (Bianchi et al., 2021) | $H^{(K)} = \sigma(\tilde{L}H^{(K-1)}W_1 + XW_2)$ | $H^{(K)} = \sum_{t=0}^K \tilde{L}X W_2^t W_1^{K-t}$ |
| APPNP (Klicpera et al., 2019) | $H^{(k+1)} = (1-\alpha)\hat{A}H^{(k)} + \alpha H^{(0)}; H^{(0)} = \sigma(XW_1)W_2$ | $H^{(K)} = \sum_{t=0}^K (1-\alpha)^t \hat{A}^t H^{(0)} + \sum_{i=0}^{t-1} \alpha(1-\alpha)^i \hat{A}^i H^{(0)}W_1W_2$ |
| ChebyNet (Defferrard et al., 2016)** | $H = \sum_{k=0}^K P_k(\tilde{L})XW^{(k)}$ | $H^{(K)} = \sum_{t=0}^K P_t(\tilde{L})XW^{(t)}$ |
| GPRGNN (Chien et al., 2021) | $H = \sum_{k=0}^K \gamma^{(k)}\tilde{L}^k\sigma(XW_1)W_2$ | $H^{(K)} = \sum_{t=0}^K \tilde{L}^t X \gamma^{(t)}W_1W_2$ |
| BernNet (He et al., 2021) | $H = \sum_{k=0}^K \frac{1}{2^K}\binom{K}{k}\gamma^{(k)}(2I - \tilde{L})^{K-k}\tilde{L}^k\sigma(XW_1)W_2$ | $H^{(K)} = \sum_{t=0}^K (2I - \tilde{L})^{K-k}\tilde{L}^t X \gamma^{(t)}W_1W_2$ |
| WGNN (Ours) | $H = \sum_{k=0}^K P_k(\tilde{L})XW^{(k)} + SW^{(s)}$ | $H = \sum_{k=0}^{K-1} P_k(\tilde{L})XW^{(t)} + \sum_{j=0}^{J-1} S_j W^{(j)}$ |

* Without specification, $H^{(0)} = X$.

** $T_k(x)$ denotes Chebyshev polynomial $P_0(x) = 1, P_1(x) = x, P_k(x) = 2xP_{k-1} - P_{k-2}$.

