# OpenReview forum: "Wide Graph Neural Network"
_ICLR.cc/2023/Conference — Submitted to ICLR 2023_

### Official Review · Reviewer_Hwom · 2022-10-21

**Confidence:** 4
**Correctness:** 4
**Technical Novelty And Significance:** 3
**Empirical Novelty And Significance:** 4
**Recommendation:** 6

**Clarity, Quality, Novelty And Reproducibility:**

Clarity: The paper is fairly well-written.

Quality: The empirical results are convincing.

Originality: On the empirical/practical side WGNNs are novel to the best of my knowledge. The theoretical aspects of the work do not seem novel to me (see the last weakness above).

**Strength And Weaknesses:**

Strengths:

In my view, the main strengths of this article are the empirics:
- the experimental validation that WGNNs achieve SOTA on Cora, Citeseer, PubMed, and a range of other standard tasks is impressive
- I found the ablation studies to be useful, especially how high order a polynomial to use and the how much of the signal to keep in the SVD.
- I found it interesting that logistic regression on top of frozen features (the polynomials P_k and the singular vectors/values of the normalized adjacency matrix) in WGNNs outperforms deep neural networks of various kinds. This fact is also kind of a weakness of this paper (see below).

Weaknesses:

In my view, the main weaknesses are in the conceptual framing, the theory, and connecting the theory to the empirics:

- I don’t think it’s really fair to call WGNNs neural networks. After all, they are just logistic regression on top of frozen features (the polynomials P_k and the singular vectors/values of the normalized adjacency matrix). As mentioned above, I found the fact that these simple models work so well to be surprising. But it is also somewhat incongruous with the rest of the paper. Namely, the way I read this paper is that actually on many graph-based tasks it is better to just use hand-crafted or at least pre-computed features. This doesn’t seem to have much relation to the discussion of GCNs and oversmoothing in the first few sections.
- Overall, the theory in this article is weak:
    - Theory is done only for linear networks
    - Theory does not cover many other kinds of commonly used architectures such as residual GNNs and things like GCNii, which provably alleviate oversmoothing.
    - No theoretical analysis directly for WGNNs is provided
    - The main theory result, Theorem 3.1, has been proved many times in various contexts. For example:
        - See appendix B.3 in Huang, Wei, et al. "Towards Deepening Graph Neural Networks: A GNTK-based Optimization Perspective." arXiv preprint arXiv:2103.03113 (2021).
        - See Lemma 3.1 in Cai, Chen, and Yusu Wang. "A note on over-smoothing for graph neural networks." arXiv preprint arXiv:2006.13318 (2020).
        - See Theorem 2 in Huang, Wenbing, et al. "Tackling over-smoothing for general graph convolutional networks." arXiv preprint arXiv:2008.09864 (2020).

**Summary Of The Paper:**

This article introduces wide graph neural networks (WGNNs). The structure of WGNNs is inspired by an analysis of oversmoothing in linear GNNs. Experiments are presented in which WGNNs achieve SOTA in several benchmark tasks, especially in the setting of heterphilic graphs.

**Summary Of The Review:**

This article proposes wide graph neural networks (WGNNs), which are linear classifiers over features given by polynomials in the graph adjacency matrix and its individual singular vectors. Somewhat surprisingly, WGNNs achieve SOTA on a range of homophilic and heterophilic benchmark tasks, achieving especially impressive gains on the heterophilic tasks. While the authors use a significantly simplified analysis of linear GNNs to motivation WGNNs, I did found their analysis to lack in novelty and not be related to WGNNs directly.

---

> ### Author Response · Authors · 2022-11-26
> **Many Thanks to and Welcome Further Discussions**
>
> Dear Reviewer Hwom,
>
> Thanks for your constructive suggestions and questions. Regarding your question points, we append further experiments/explanations, most of which help improve our paper. We are eager to discuss the raised questions to ensure they are well addressed, or there still exist unclear points. Please don’t hesitate to share any ideas!
>
> Many thanks,
> Paper 3063 Authors

---

### Official Review · Reviewer_HybF · 2022-10-24

**Confidence:** 4
**Correctness:** 3
**Technical Novelty And Significance:** 2
**Empirical Novelty And Significance:** 2
**Recommendation:** 3

**Clarity, Quality, Novelty And Reproducibility:**

- Overall, most parts of the paper are well-written while some notations are ambiguous. The presentation of experimental results is satisfactory.

- The theoretical analysis and conclusion are not impressive. The motivation of the final term in WGNN is unclear.

- The details of experiments are provided.

**Strength And Weaknesses:**

### Strength

- The paper is well-organized and easy to follow.
- The experimental results are satisfactory, especially on heterophilic graphs. The presentation of experiments is quite good.

### Weakness

- The theoretical analysis and conclusion provided in this paper seem not new. Some existing works have rigorously investigated the over-smoothing caused by high-order graph operations, including but not limited to [1] [2].
- The discussion of "poor flexibility" of spectral GNNs is unconvincing. The authors claimed that the poor flexibility is caused by the limited size of parameters for $P_k(L)$. One can increase the number of these parameters, $\gamma$, similar to GCN (proposed by Kipf&Welling). I agree that both spatial and spectral methods cannot handle the heterophilic cases, but the poor flexibility of spectral methods may be unconvincing in my opinion.
- How is the time cost per epoch computed? Does it contain the time to compute $\Phi_t$? Since the insufficiency problem of GNNs is actually caused by the graph operations, it may be unconvincing to only report the time of each epoch during BP.
- I'm not persuaded by the motivation of $S_j$. Why is the "dimensionality reduction" needed for GNNs?

(Minor)

- In Figure 1-middle, the authors may misuse $\gamma$, which seems to be $\theta$ in the top formulation.
- At the bottom of Page-3, the authors claim "a GNN model forms *K* feature sub-spaces". Does $K$ mean $T$? After reading Sections 2 and 3, $K/k$ and $T/t$ are pretty confusing and ambiguous though I may get the meaning in most cases.

[1] Kenta Oono and Taiji Suzuki, Graph neural networks exponentially lose expressive power for node classification, ICLR, 2020.
[2] Weilin Cong, Morteza Ramezani, and Mehrdad Mahdavi, On provable benefits of depth in training graph convolutional networks, NeurIPS, 2021.

**Summary Of The Paper:**

This paper propose a new GNN model, namely wide graph neural network (WGNN), to address the problems existing in spatial and spectral GNNs, including *over-smoothing*, *homophily and heterophily*, and *poor flexibility*.

**Summary Of The Review:**

I'd like to update my score during the discussion period if my concerns could be well addressed.

---

> ### Author Response · Authors · 2022-11-26
> **Many Thanks to and Welcome Further Discussions**
>
> Dear Reviewer HybF,
>
> Thanks for your huge efforts in reviewing our paper. Regarding your question points, we append further experiments/explanations, most of which help improve our article. We are eager to discuss the raised questions further. Please let us know if you still have any unclear parts of our work.
>
> Many thanks,
> Paper 3063 Authors

---

### Official Review · Reviewer_KH5p · 2022-10-24

**Confidence:** 4
**Correctness:** 1
**Technical Novelty And Significance:** 1
**Empirical Novelty And Significance:** 1
**Recommendation:** 1

**Clarity, Quality, Novelty And Reproducibility:**

- Clarity: the paper is clear and easy to read.
- Quality: the paper contains several mistakes and is based on false premises.
- Novelty: the main novel contributions claimed by the paper depend on a misrepresentation of the literature. The novelty of the overall work is very low.
- Reproducibility: I tried and failed to reproduce the reported results using the code provided by the authors.

**Strength And Weaknesses:**

**Strengths**:
- The paper is easy to read.
- The authors report good performance on five benchmarks with homophilic graphs and 2 benchmarks with heterophilic graphs (although please see below for reproducibility issues).

**Weaknesses**:
- The definition of spectral GNNs given by the authors is incorrect, since not all spectral GNNs need to eigendecompose the Laplacian and not all of them use polynomial filters. Also, Equations (4) and (8) do not describe the typical polynomial filter. This is particularly concerning since this definition (and its supposed limitations) is one of the main premises of the paper, which is therefore built on false assumptions.
  I would also point out that GCN is not significantly different from ChebyNets, so they are definitely not representatives of two different classes of GNNs.
- Oversmoothing is a problem that has been studied very in depth in GNN literature, to the point where it can be considered essentially solved. People have been training very deep GNNs for years now, using techniques like skip connections, adaptive information propagation, edge/node drop, and weight tying.
- The authors claim that solving oversmoothing is still needed to train GNNs hundreds of layers deep, but then only test models with up to 10 layers (which, as I said above, is not a complicated feat to achieve).
- Since, by construction, Equation 5 only works for non-parametric transformations of the graph, all GNNs that have learnable edge-dependent message functions cannot be modeled by the proposed method. Some of these, like Graph Attention Networks and Edge-conditioned Convolutions, are quite significant. This limitation is not due to "nonlinearity" (as claimed in the conclusions).
  Also, Equation 5 is far from a unification of spectral and spatial methods, but rather a compact way of writing polynomial filters.
- The main proposal of Equation (9) is to use a polynomial filter and to add a term consisting of learnable linear projections of the first few eigenvectors of the adjacency matrix, which is both not novel and very expensive (quadratic in the number of nodes).
- While one of the main goals of the paper is to solve oversmoothing, the experiments only show minor improvements on very standard datasets (Cora, Citeser, and Pubmed) with no concrete study of oversmoothing.
- There is no reason why the proposed design would, a priori, have better performance on heterophilic graphs. The theoretical analysis on page 6 also does not justify why the proposed method should work better.  The method is still essentially a polynomial filter, so I don't see why it would outperform all other methods by this much on Squirrel and Chameleon.
  For this reason, I took the liberty of running the code provided in the supplementary material by the authors, using the `--poly` flag to test different methods. As expected, all methods I have tried (GPR, Cheby, GCN, and the authors') achieve an accuracy above 73 on Chameleon. Please let me know if I have misused the code in some way or if I misunderstood something.

**Summary Of The Paper:**

This paper presents Wide Graph Neural Networks, a class of polynomial graph neural networks (GNNs) with an additional learnable term acting on the principal eigenvectors of the adjacency matrix.

The paper attempts to study and unify the literature on GNNs to highlight the limitations of previous methods, aiming to solve the issue of oversmoothing and to improve classification performance on heterophilic graphs.

**Summary Of The Review:**

The paper has significant flaws and contains mistakes, the novelty is very limited, and the results do not seem to be reproducible (using the authors' code).

I recommend the paper is rejected.

---

> ### Author Response · Authors · 2022-11-26
> **Many Thanks to and Welcome Further Discussions**
>
> Dear Reviewer KH6p,
>
> Thanks for your precise time and efforts in reviewing our paper. Regarding your question points, we append further experiments/explanations, most of which help improve our article. We are eager to discuss the raised questions further. Please let us know if you still have any unclear parts of our work.
>
> Many thanks,
> Paper 3063 Authors

---

### Official Review · Reviewer_Uh1P · 2022-11-03

**Confidence:** 3
**Correctness:** 3
**Technical Novelty And Significance:** 3
**Empirical Novelty And Significance:** 3
**Recommendation:** 6

**Clarity, Quality, Novelty And Reproducibility:**

This paper is clearly written. The proposed WGNN method is novel. The authors also provide the code used in the experiments as supplementary material.

**Strength And Weaknesses:**

Pros:

1. The proposed Wide Graph Neural Network (WGNN) is novel. It is an interesting idea to decompose the components in spectral and spatial GNNs into a non-parametric feature space and a parameter space to re-weight the corresponding feature subspace.

2. The authors provide detailed theoretical analysis to explain over-smoothing and poor performance on heterophilic graphs. Unlike spectral GNNs, which use a single parameter matrix for all polynomial terms, WGNN has better flexibility by allowing different parameters for each term.

3. Experimental results on eight datasets (including both homophilic and heterophilic) show that the proposed WGNN outperforms several baselines in terms of node classification accuracy.

Cons:

1. In the ablation study, the nearly optimal value for the order K of the polynomials is 3 (based on the Cora and Chameleon datasets). This value is also the number of feature sub-spaces. I find it a bit difficult to understand that the optimal number of feature subspaces in WGNN could be a fixed value for different types of graph data and applications.

2. It would be great if the authors can provide more explanations on why principal components of the adjacency matrix (i.e., low dimensional information for the graph structure) is used to form the subspace $S_j$. As mentioned in the paper, $S_j$ could be any transformation of the adjacency matrix. I'm wondering the classification performance with other types of transformations.

3. What is the total number of parameters in WGNN, comparing with spectral and spatial GNNs? As discussed in the conclusion, the parameters in WGNN can be further reduced by introducing additional constraints and assumptions.

4. Among all the datasets, WGNN does not outperform GNN-LF and GNN-HF on CiteSeer data. It would be great if the authors can provide some insights on this.

**Summary Of The Paper:**

The authors propose a unified view for both spectral and spatial GNNs from the matrix space analysis perspective to investigate possible reasons for over-smoothing, poor flexibility, and low performance on heterophily. They propose a new GNN framework, namely, Wide Graph Neural Network (WGNN), to address these issues. WGNN consists of two components: one is for constructing a non-parametric feature space, and the other is for learning the parameters to re-weight the feature space.

**Summary Of The Review:**

The proposed Wide Graph Neural Network (WGNN) is novel. Experimental results on several graph datasets show the superiority of WGNN comparing with the baselines. The authors provide detailed theoretical analysis on how WGNN can help address the issues of over-smoothing, poor flexibility, and low performance on heterophilic data. I would also like to see the authors' response to my questions
raised in the cons above during the rebuttal period.

---

> ### Author Response · Authors · 2022-11-26
> **Many Thanks to and Welcome Further Discussions**
>
> Dear Reviewer Uh1P,
>
> Thanks for your constructive suggestions and questions. Regarding your question points, we append further experiments/explanations, most of which help improve our paper. We are eager to discuss the raised questions to ensure they are well addressed, or there still exist unclear points. Please don’t hesitate to share any ideas!
>
> Many thanks,
> Paper 3063 Authors

---

### Author Response · Authors · 2022-11-26
**General Responses to ACs and Reviewers for the First Round Rebuttal**

Dear area chairs and reviewers,

We, the authors, sincerely appreciate all the reviewers’ tremendous efforts in reviewing our script. The provided suggestions and raised questions are promoting and greatly help us improve our work! Please allow us to clarify our work's contributions, which remain unchanged throughout the rebuttal phase.

1. We decompose the components in GNNs into a non-parametric feature space and a parameter space to re-weight the corresponding feature subspace. This
2. Based on this decomposition, we provide a detailed theoretical analysis to explain over-smoothing and poor performance on heterophilic graphs. We propose a wide graph neural network (WGNN) to alleviate these issues.
3. Experimental results on both homophilic and heterophilic datasets show that WGNN outperforms several baselines regarding node classification accuracy.

However, based on the feedback from the reviewers, we mainly find three blurred parts in our script due to insufficient narrative or limited comparisons of some related work:

1. More related analytical works should be discussed in the theoretical analysis part.
2. Experiments against over-smoothing are somehow limited.
3. More variants of WGNN should be compared to demonstrate this framework's effectiveness better.

Regarding the suggestions, in the revised version, we append the following content:

1. Specified concepts and more discussions of related work are included in Sec. 2 and Appendix B.3.
2. More extensive experiments on over-smoothing are appended in Appendix C.6, e.g., comparing much deeper layers/orders and other related proposals.
3. We conduct further experiments on weight-sharing schemes, other transformations for dimension reduction, and hyper-parameters selection in Appendix C.4, C.5, and C.7.

If it is necessary, we will take the revised parts into the main text. Except for these changes, we respond to the remained questions below for each reviewer.

Hopefully, our responses can address most of the confusion. Please don’t hesitate to raise further questions if there is any inconsistent understanding.


Many thanks,
Paper 3063 Authors

---

### Decision · Program_Chairs · 2023-01-20

**Decision:**

Reject

**Justification For Why Not Higher Score:**

There have been split scores for this paper, but ultimately the reviewers largely agreed about the key concerns around this paper. Essentially, for a new, unified view of the GNN framework it would be important to understand what is making it work better compared to the previous approaches. However, the reviewers remain unconvinced about the provided arguments, which cannot explain their surprise to the demonstrated results. I believe that addressing this issue as suggested is needed, in order to allow other researchers better understand how to build upon this work.

**Justification For Why Not Lower Score:**

N/A

**Metareview: Summary, Strengths And Weaknesses:**

This work is presenting a unifying view of GNNs named Wide Graph Neural Network (WGNN). The key idea is to view graph learning under the lens of dictionary learning, aiming to investigate and eventually tackle typical problems associated with GNNs, namely oversmoothing, poor flexibility and low performance on heterophilic graphs.

Four expert reviewers provided thorough reviews about this paper. I note that, in addition to the visible author-reviewer discussions, for this paper there has also been discussion among reviewers and the AC both on and off openreview. I also note that a score of 1 given by one of the reviewers is too harsh for this paper, and I have treated it as a higher score.

Overall, this paper is easy to follow. They key idea is found by the reviewers to be interesting. The experimental evaluation is thorough, especially taking into account the additional results provided during the rebuttal period. Further, the actual obtained quantitative results are deemed good by the reviewers.

However, although the authors offer a new unified view, the actual mathematical framework still does not seem very novel (see detailed comments by reviewer HybF and KH5p) and the connection with other methods attacking oversmoothing is still unclear (see comments by reviewer Hwom). Further, there is still confusion about the generality of the assumptions, e.g. equation (4). This is important not because there is necessarily an insistence on high novelty, but because this is a new view of an established class of models so it should be very well connected to the literature.

With the above in mind, a second related issue is that the results are somehow surprising: the theoretical analysis (limited to the linear case) does not explain the counter-intuitive fact that a polynomial filter can achieve such results. This is not to say that the experimental results are wrong, but that it would be essential for the reader to be convinced also from the theoretical perspective why this is the case, and I agree with the reviewer who suggests to tie the empirics with the theory.

Despite a very helpful rebuttal by the authors, the above issues seem to be major enough that a separate submission is needed to address it, as the reviewers have not been convinced to raise their score. As this might indeed be a presentation issue, the reviewers offered two related suggestions to improve the manuscript: firstly, to further expand the theoretical analysis with convincing arguments as to why WGNN improves upon previous methods that utilize the same key ingredients such as polynomial filters and what is missing from previous works that cannot achieve the same results (and then carefully ablate these parts). Secondly, to tie the empirical evaluation to the theory more convincingly. This is essential for the community to understand what are the key ingredients contributing to the results, such that further research can more easily be built upon.